# Quantitative models reveal the organization of diverse cognitive functions in the brain

Tomoya Nakai[1,2] & Shinji Nishimoto [1,2,3✉]

Our daily life is realized by the complex orchestrations of diverse brain functions, including perception, decision-making, and action. The essential goal of cognitive neuroscience is to reveal the complete representations underlying these functions. Recent studies have characterised perceptual experiences using encoding models. However, few attempts have been made to build a quantitative model describing the cortical organization of multiple active, cognitive processes. Here, we measure brain activity using fMRI, while subjects perform 103 cognitive tasks, and examine cortical representations with two voxel-wise encoding models. A sparse task-type model reveals a hierarchical organization of cognitive tasks, together with their representation in cognitive space and cortical mapping. A cognitive factor model utilizing continuous, metadata-based intermediate features predicts brain activity and decodes tasks, even under novel conditions. Collectively, our results show the usability of quantitative models of cognitive processes, thus providing a framework for the comprehensive cortical organization of human cognition.

[1] Center for Information and Neural Networks (CiNet), National Institute of Information and Communications Technology (NICT), Osaka, Japan. [2] Graduate School of Frontier Biosciences, Osaka University, Osaka, Japan. [3] Graduate School of Medicine, Osaka University, Osaka, Japan. ✉email: nishimoto@nict.go.jp

The cortical basis of daily cognitive processes has been studied using voxel-wise encoding and decoding modeling approaches[1]. These approaches employ multivariate regression analysis to determine how the brain activity in each voxel is modeled by target factors, including visual features[2,3], object or scene categories[4–6], sound features[7–9], and linguistic information[10–12]. Moreover, other studies have further described the cortical (e.g., semantic) representational space that elucidates important categorical dimensions in the brain (e.g., mobile vs. nonmobile, animate vs. inanimate) and how such representations are mapped onto the cortex[4,10]. However, the previous studies have relied on brain activity recorded during passive listening or viewing tasks, and no study has yet clarified the comprehensive cortical representations underlying active cognitive processes

In this study, we address this issue by combining encoding modeling and metadata-based reverse-inference to further reveal such representations. Six subjects underwent functional MRI experiments to measure whole-brain blood-oxygen-level-dependent (BOLD) responses while they performed 103 naturalistic tasks (Fig. 1a, b). The tasks included as many cognitive varieties as possible and ranged from simple visual detection to complex cognitive tasks, including memorization, language comprehension, and calculation (see Supplementary Methods for the task list and associated descriptions). Our experimental setup is aimed at extending previous efforts at describing the semantic space[4,10] by estimating the cognitive space that depicts the relationships among diverse cognitive processes. In this process, each task was regarded as a sample taken from the entire cognitive space. To obtain a comprehensive representation of the cognitive space, we modeled voxel-wise responses using regularized linear regression[1] based on two sets of features (Fig. 1c, d). First, using a task-type encoding model, in which tasks are represented as binary labels (Fig. 1c), we evaluated the representational relationships among the cognitive tasks across the cerebral cortex. Second, to further examine the generalizability of the modeling approach to the cognitive tasks, we constructed an additional cognitive factor encoding model, in which each task was transformed into a 715-dimensional continuous feature space using metadata references[13] (Fig. 1d). This enabled the latent feature space to be used for each task[1,14]. Furthermore, we were able to predict and decode activity for novel tasks not used during model training (Fig. 1e). Our framework provides a powerful step toward the comprehensive modeling of the brain representations underlying human cognition.

## Results

**Hierarchical organization of cognitive tasks**. To examine how the cortical representations of over 100 tasks were related, we modeled task-evoked brain activity using a task-type model (Fig. 1) and calculated a representational similarity matrix (RSM) using the estimated weights, which were concatenated across six subjects (Fig. 2a). The training dataset consisted of 3336 samples (6672s) and the test dataset consisted of 412 samples (824s, repeated four times). The representational relationship of over 100 tasks was further visualized by a dendrogram obtained using hierarchical clustering analysis (HCA). The HCA results suggested that the tasks formed six clusters based on their associated representational patterns in the cerebral cortex. The largest clusters contained tasks based on sensory modalities, such as visual ("AnimalPhoto," "MapSymbol"), auditory ("RateNoisy," "EmotionVoice"), and motor ("PressLeft," "EyeBlink") tasks. In addition, some clusters contained higher cognitive components, such as language ("WordMeaning," "RatePoem"), introspection ("ImagineFuture," "RecallPast"), and memory ("MemoryLetter," "RecallTaskEasy"). Although six clusters were determined by

visual inspection for a descriptive purpose, the same analyses can be performed on any subclusters in the dendrogram. We also obtained a similar hierarchy pattern using an RSM calculated directly from brain activity (see Supplementary Fig. 1 and Supplementary Note 1).

To investigate the plausible cognitive factors related to each task cluster, we next performed a metadata-based evaluation of the cognitive factors. For each of the cortical maps of the task cluster weight matrix (Supplementary Fig. 2 and Supplementary Note 2), we calculated Pearson's correlation coefficients with the 715 reverse-inference maps taken from the Neurosynth database[13]. The top ten terms for the most task clusters were consistent with our interpretation based on the included task types (Table 1). High correlation was observed between the visual cluster and vision-related terms (e.g., "visual", "perceptual"), the memory cluster and working memory-related terms ("working memory", "executive"), and the language cluster and language-related terms ("language", "reading"). In addition, the motor cluster showed a high correlation with motor-related terms ("movement", "motor"), and the auditory task cluster showed a high correlation with auditory-related terms ("auditory", "listening"). Moreover, the introspection cluster showed a high correlation with the default mode-related terms ("default mode", "default network"). The reverse-inference analysis is applicable to any subcluster (Supplementary Table 1). For example, there is a time-perception subcluster ("TimeSound", "Rhythm") within the Auditory cluster. The reverse-inference analysis assigned cognitive factors of "timing", "monitoring", and "working memory" to this subcluster in addition to the auditory factors. In the knowledge-recalling subcluster ("RecallKnowledge," "Category-Fluency") within the Introspection cluster, the reverse-inference analysis assigned cognitive factors of "phonological", "production", and "language" even though the participants were not asked to overtly produce linguistic information. These results suggested that data-driven reverse-inference is effective in providing a valid interpretation of the cognitive factors underlying the different task clusters.

The tasks were further represented in the subclusters of the specific cognitive properties (Fig. 2b–d). For example, in the visual cluster, tasks with food pictures ("RateDeliciousPic", "DecideFood") were closely located (Fig. 2b), whereas tasks with negative pictures ("RateDisgustPic", "RatePainfulPic") formed a separate cluster. Memory tasks involving calculations ("CalcEasy", "CalcHard") were closely located (Fig. 2c), whereas those involving simple digit matching ("MemoryDigit", "MatchDigit") formed a separate cluster. For the introspection cluster (Fig. 2d), tasks involving the imagination of the future and recalling of past events ("ImagineFuture," "RecallPast") were more closely located than those involving the imagination of places or faces ("ImaginePlace", "RecallFace").

To further explore whether hierarchical information is useful in capturing cortical representation, we constructed an additional hierarchical model that was based on the task clusters subordinated by each non-terminal node in the dendrogram (see Methods section). By comparing the prediction accuracy of brain activity using the task-type model (Fig. 1c) and the hierarchical model, we found that the hierarchical model outperformed the task-type model (hierarchical model, mean ± SD, 0.313 ± 0.046; task-type model, 0.293 ± 0.053; one-sided Wilcoxon signed-rank tests, $p < 0.001$ for all subjects). These results indicated that the hierarchically organized brain representations of cognitive tasks can be captured using this modeling procedure.

**Visualization of cognitive space and its cortical mapping**. HCA revealed the relative relationships between the task samples

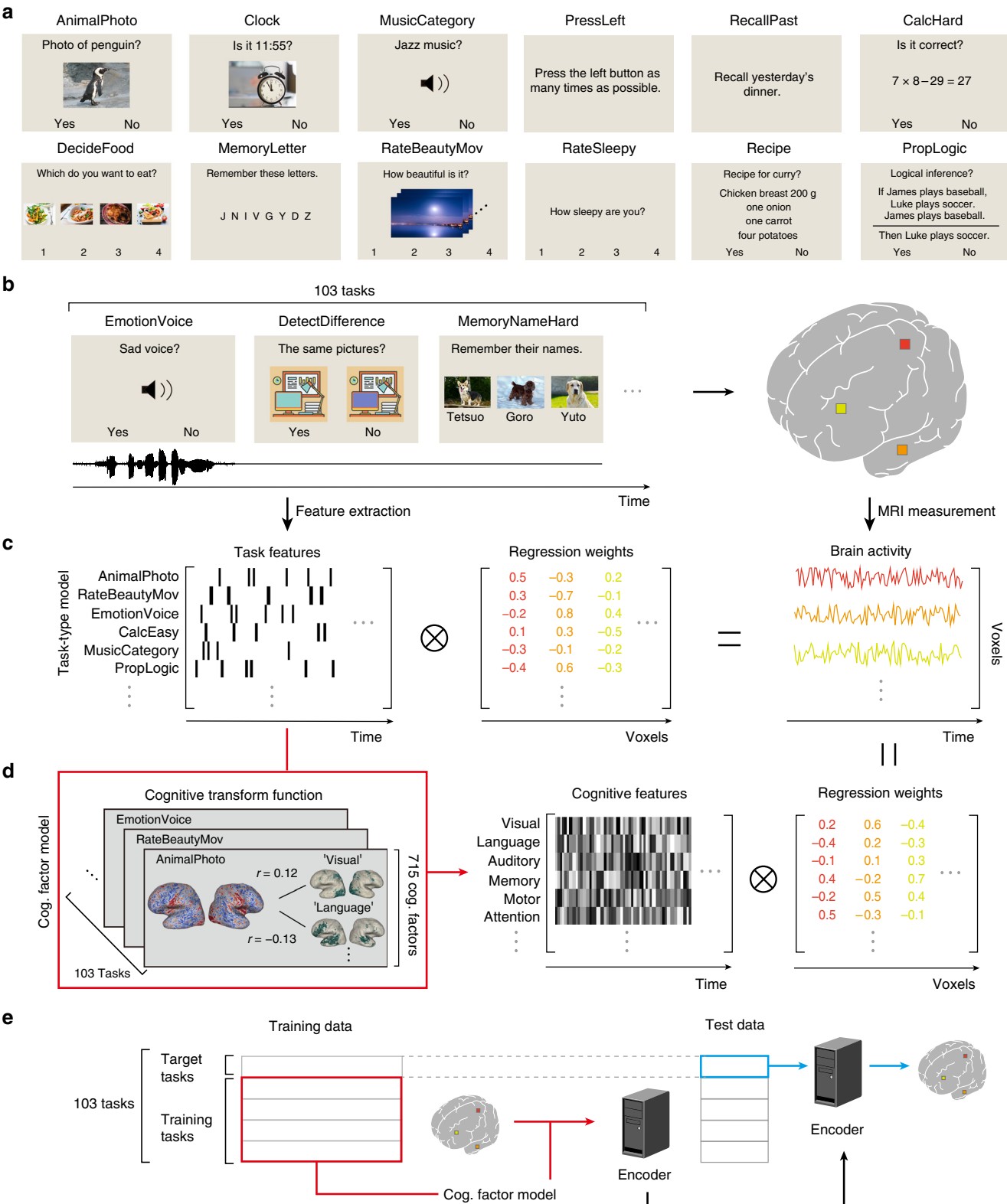

**Fig. 1 Schematic diagrams of the task setting and analysis methods. a** Example image of 12 tasks, with task names described at the top. **b** The subjects performed 103 naturalistic tasks while the brain activity was measured using functional MRI. **c** Schematic of the encoding model fitting using the task-type model. **d** Schematic of the cognitive factor model. The cognitive transform function was calculated based on correlation coefficients between the weight maps of each task and the 715 metadata references[13]. Task-type features were transformed into cognitive factor features. **e** Schematic of the encoding model fitting using the cognitive factor model for novel tasks. Target tasks were not included in the model training datasets (in red). The trained encoder provided a prediction of brain activity (in blue). Note that some visual images used in the tasks are different from the original because of copyright protection. Penguin, clock, food, sky, desktop illustration, and dog images are provided by Storyblocks.

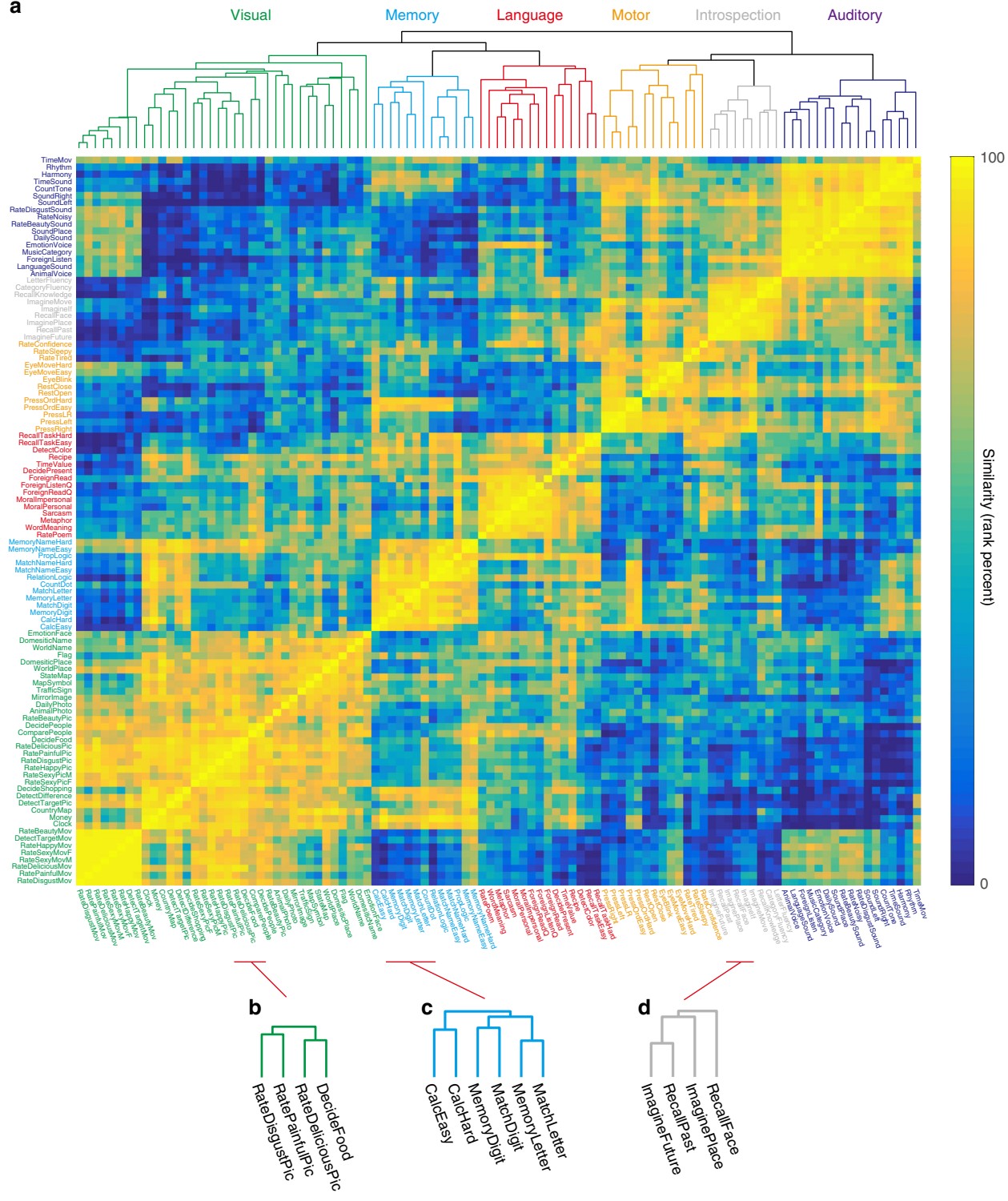

**Fig. 2 Hierarchical organization of over 100 tasks. a** Representational similarity matrix of the 103 tasks (see Supplementary Methods for the descriptions of each task), reordered according to the hierarchical cluster analysis (HCA) using task-type model weights (concatenated across subjects). The dendrogram shown at the top panel represents the results of the HCA. The six largest clusters were named after the included task types. **b–d** Example task subclusters and their dendrograms in the visual (**b**), memory (**c**), and introspection (**d**) clusters.

taken from the entire cognitive space. To further determine the structure and cortical organization of the cognitive space, we performed principal component analysis (PCA) using the estimated weight matrix of the task-type model, concatenated across subjects (Fig. 3a). Accordingly, Fig. 3a shows the distributions of the tasks according to their PCA loadings, where

the task position was determined by the first and second PCs, with task color determined by the first, second, and third PCs (corresponding to red, green, and blue, respectively; see Fig. 3a inset). Tasks with similar representations were assigned similar colors and closely located within the two-dimensional space. Tasks involving movie processing were clustered on the top left,

**Table 1 Top cognitive factors related to each task cluster.**

| | Top cognitive factors in the neurosynth database |
|---|---|
| Visual cluster | "visual", "object", "face", "motion", "viewing", "perceptual", "vision", "visual stream", "sighted", "recognition" |
| Memory cluster | "working memory", "task", "calculation", "load", "attentional", "numerical", "spatial", "arithmetic", "subtraction", "executive" |
| Language cluster | "reading", "language", "comprehension", "sentence", "semantic", "word", "linguistic," "native," "syntactic", "lexical" |
| Motor cluster | "finger," "motor", "hand", "sensorimotor", "somatosensory", "movement", "motor imagery", "execution", "tactile", "tapping" |
| Introspection cluster | "default mode", "autobiographical", "default network", "self-referential", "resting state", "episodic", "theory mind", "retrieval" "personal", "mentalizing" |
| Auditory cluster | "auditory", "sound", "pitch", "listening", "acoustic", "speech", "music", "audiovisual", "hearing", "vocal" |

Top 10 cognitive factors (excluding similar terms) in the Neurosynth database for each of the six task clusters, based on the correlation coefficients between the task weight map and the 715 registered reverse-inference maps.

whereas those dedicated to image and auditory processing were located more centrally on both the left and right sides, with a gradual shift toward complex cognitive tasks involving language, memory, logic, and calculation at the bottom of the distribution. We also obtained a similar distribution using a multi-dimensional scaling directly applied to brain activity (Supplementary Fig. 3 and Supplementary Note 3).

To interpret each PC, we first determined those that were meaningful based on the explained variance (Fig. 3b). The top four PCs that explained more than 5% of the variance were regarded as meaningful, and the dominant cognitive components were further analyzed based on these PCs. We then quantified the relative contribution of each task using PCA loadings; tasks with higher PCA loading values were contributing more to the target PC (Supplementary Table 2 and Supplementary Note 4). Thus, each PC was labeled based on these cognitive tasks. To obtain an objective interpretation of the PC labeling, we also performed a metadata-based inference of the PC-related cognitive factors (Supplementary Table 3).

To further examine the relationship between HCA and PCA results, we next calculated the relative contribution of the top four PCs to the six largest task clusters (Fig. 3c). By averaging the PCA loadings of the tasks included in each of the target clusters, we found that the top PCs corresponded well to the related clusters, with mean PCA loadings significantly different from zero (two-sided sign test, $p < 0.05$, FDR-corrected). PC1 (the auditory component) contributed to the auditory cluster; PC2 (the audiovisual component) contributed to the auditory and visual clusters; PC3 (the language component) contributed to the language cluster; and PC4 (the introspection component) contributed to the motor and introspection clusters. These results revealed the representational correspondence between the HCA and PCA results.

To further visualize the cortical distributions of the cognitive space representations, voxel-wise PCs were projected to the cortical sheet of each subject (Fig. 4a, Supplementary Figs. 4, 5 and Supplementary Note 5) using the same RGB color scheme as shown in Fig. 3a. For example, the occipital areas were presented mostly in green, showing that voxels in these areas signify movie-related and image-related tasks (Fig. 3a). The frontal areas showed intricate patterns, including language-related representations (blue) in the left lateral regions. This topographical organization was consistent across subjects (Supplementary Fig. 5), indicating that our analyses provided a broad representation of the cognitive space in the human cerebral cortex.

To examine how each cortical voxel differs in its representation of over 100 tasks (task selectivity), we visualized voxel-wise task weights on the two-dimensional cognitive space depicted in Fig. 3a. We found a representation of language-related tasks in the middle temporal voxel (Fig. 4b), introspection-related tasks in the left medial frontal voxel (Fig. 4c), and auditory-related tasks in the right superior temporal voxel (Fig. 4d).

Using this visualization method, scrutiny of fine mapping of task selectivity of any cortical voxel is possible. For example, when we examined the left inferior parietal lobule (IPL), we found a topographic change of task selectivity along with the inferior to superior direction (Fig. 4e–h). Calculation and logical inference tasks were represented in all three voxels (Fig. 4f–h), whereas motor tasks were largely represented in the inferior voxel (Fig. 4f), and visual tasks were largely represented in the superior voxel (Fig. 4h). The middle voxel showed an intermediate representation for both motor and visual tasks (Fig. 4g). Such topographic changes of task selectivity are indicated by the cortical map (Fig. 4a, e), which displays a color change from red (inferior voxel) to black (middle voxel) to green (superior voxel). These results suggest that voxel-wise modeling can entangle the complex topography of multiple cognitive dimensions in the association cortex.

**Prediction of brain activity during novel tasks.** Although the task-type model revealed the distinctive relationships among the tasks, it is too sparse to encompass the latent and continuous features and is not generalizable to novel tasks. To address these issues, we transformed over 100 tasks into the 715-dimensional latent feature space and constructed a voxel-wise cognitive factor model (Fig. 1d). The latent feature space was obtained based on the 715 terms and their reverse-inference maps from the Neurosynth database[13]. To produce the cognitive transform function (CTF) for each subject, we calculated the correlation coefficients between the weight map for each task in the task-type model and the reverse-inference map. We then obtained the feature matrix of the cognitive factor model by multiplying the CTF by the feature matrix of the task-type model.

To examine the generalizability of this model under novel task conditions (i.e., on a task that was not used to train the model), we trained the cognitive factor model with 80% of the tasks (approximately 82 or 83 tasks) and predicted the brain activity for the remaining 20% (Fig. 1e). In doing so, the CTF was estimated using data excluding those of the target subject, assuring the generalizability to novel tasks for each subject. Our results indicated that the model achieved significant prediction accuracy throughout the entire cortex (Fig. 5 and Supplementary Figs. 6, 7; mean ± SD, 0.322 ± 0.042; 86.2 ± 5.1% of voxels were significant; $p < 0.05$, false discovery rate (FDR)-corrected).

To confirm that these results could not be merely explained by simple sensorimotor effects, we performed an additional encoding model analysis that regressed out the relevant visual, auditory, and motor components (see Methods section). This additional analysis revealed significant prediction accuracy across the cerebral cortex (mean ± SD, 0.285 ± 0.035; 82.4 ± 4.9% of voxels were significant; $p < 0.05$, FDR-corrected; Supplementary Fig. 8), indicating that the generalizability of the cognitive factor model was due to higher-order (i.e., not sensory) cognitive components.

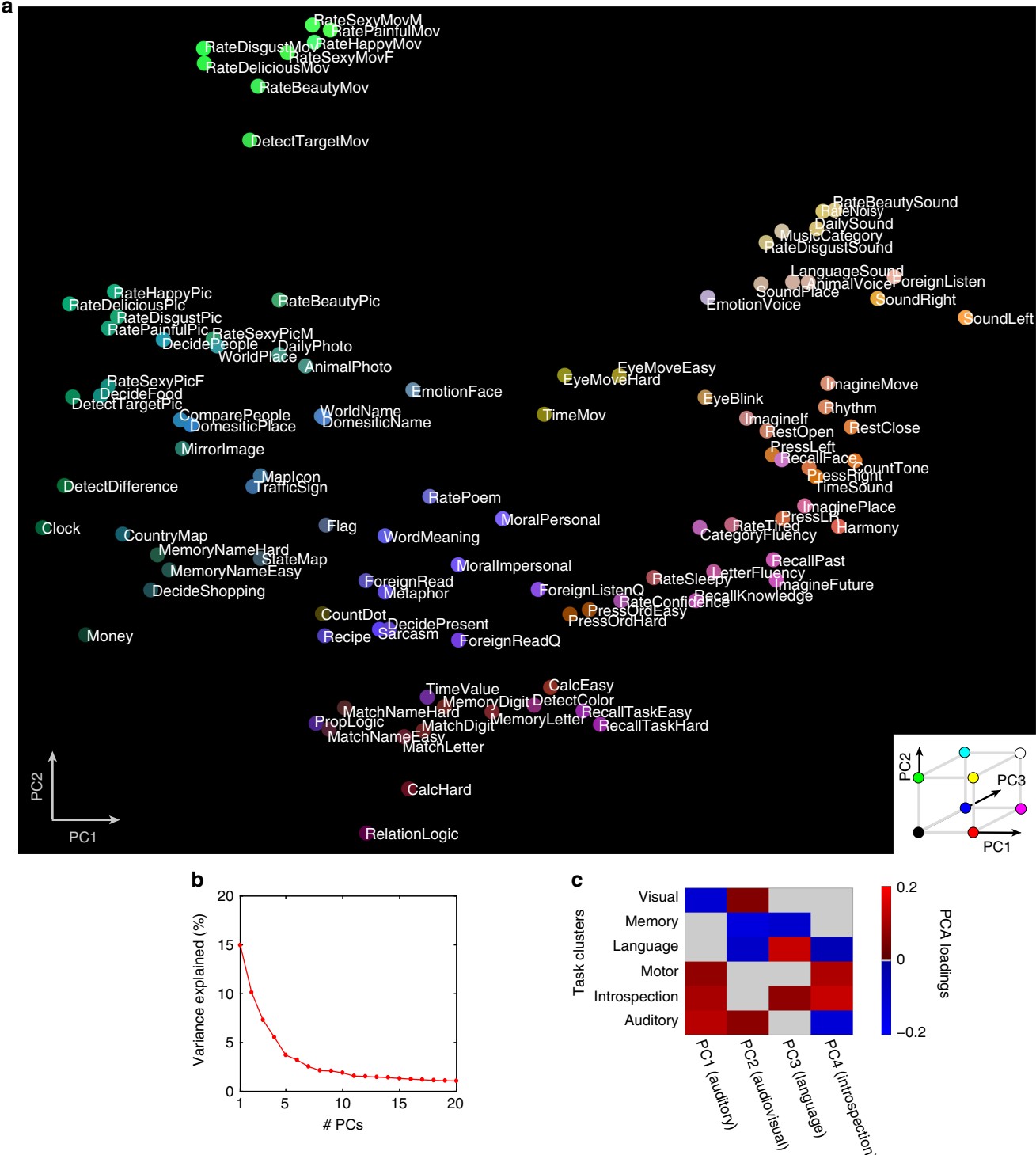

**Fig. 3 Cognitive space and cortical mapping. a** Color and spatial visualization of the cognitive space. Colors indicate the loadings of the top three principal components [PC1 = auditory (red); PC2 = audiovisual (green); PC3 = language (blue)] of the task-type model weights (concatenated across subjects), mapped onto the two-dimensional cognitive space based on the loadings of PC1 and PC2. All tasks are presented in white (see Supplementary Methods for the descriptions of each task). **b** Variance explained in the PCA. The explained variance of the original weight matrix of the task-type model was plotted for each PC. Note that the explained variance was common for all subjects in the group PCA. **c** The mean PCA loadings of the tasks in the six largest clusters, plotted for each of the top four PCs. The task clusters with positive and negative PCA loadings are shown in red and blue, respectively (significantly different from zero; two-sided sign tests, $p < 0.05$, FDR-corrected). Task clusters with nonsignificant PCA loadings are shown in gray.

**Decoding novel tasks using the cognitive factor model**. To further assess the generalizability and task specificity of the relevant cognitive factors, we performed task decoding analyses using novel tasks. To this end, we trained a decoding model to estimate the cognitive features at each time point using brain activity with hemodynamic temporal delays (Fig. 6a) for 80% of the tasks. We then quantified the task likelihood at each time point in the remaining 20% of the tasks by taking the correlation

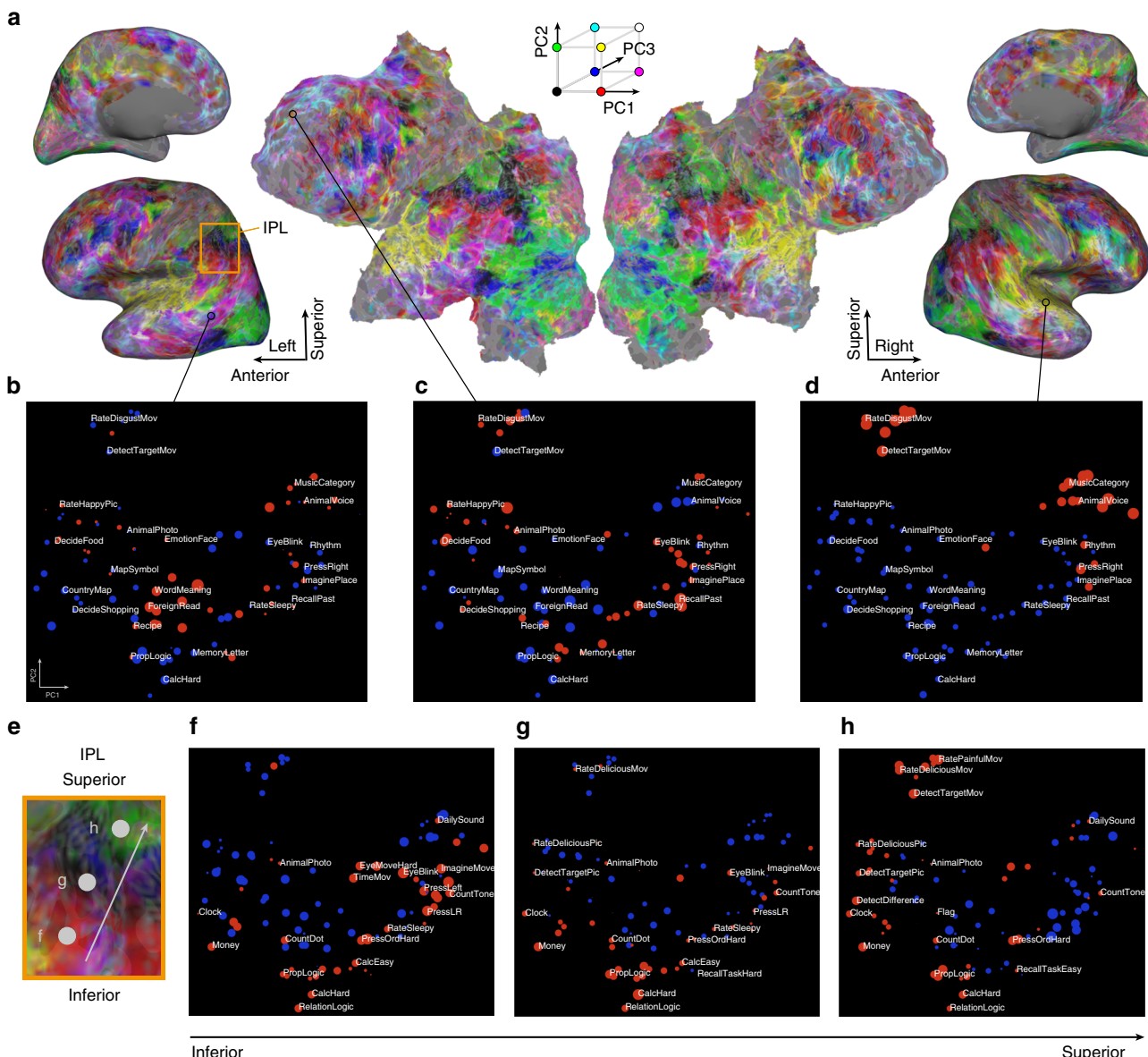

**Fig. 4 Cognitive space and cortical mapping. a** The cortical map of the cognitive space shown on the inflated and flattened cortical sheets of subject ID01 (Supplementary Fig. 5 shows all other subjects); PC1–PC3 are shown in red, green, and blue, respectively. **b–d** Examples of the task selectivity for voxels in the **b** left middle temporal, **c** left medial frontal, and **d** right superior temporal regions, mapped onto the two-dimensional cognitive space based on the loadings of the first and second principal components (PC1 and PC2, respectively) of the model weights. Tasks with positive and negative weight values were denoted in red and blue, respectively. The circle size was modulated based on the absolute weight value. For better visibility, only 24 tasks are shown in white. **e** The left inferior parietal lobule (IPL) corresponding to the orange square part in the inflated cortical sheet (**a**). **f–h** Three voxels along with the inferior to superior direction were selected to show the topographic change of task selectivity. For better visibility, only 18 tasks with positive weight values are shown in white.

between the estimated cognitive features and the template cognitive features for all 103 tasks, which was estimated by excluding the data from the target subject. We then tested whether the task likelihood for the actual (target) task was higher than that for each of the remaining 102 tasks. We obtained a significant decoding accuracy for the novel tasks (mean ± SD, 96.0 ± 0.8%; 99.5 ± 0.5% of the tasks were significant; one-sided sign tests, $p <$ 0.05, FDR-corrected; Fig. 6b, Supplementary Fig. 9), indicating that brain activity patterns were task-specific and that the portion of the human cognitive space our model covers was sufficient for decoding novel tasks.

## Discussion

Majority of previous studies using encoding or decoding model approaches used passive viewing or listening tasks[3,4,10,12]. In addition, standard neuroimaging studies using more active tasks typically focused on just a few conditions while examining the effects of pre-assumed cognitive factors by comparing induced brain activity. While the latter strategy can be a powerful way to test the plausibility of certain hypotheses, the outcomes from such specialized studies have so far not been able to elucidate the representational relationships among diverse tasks. In addition, this method cannot be generalized to naturalistic tasks where

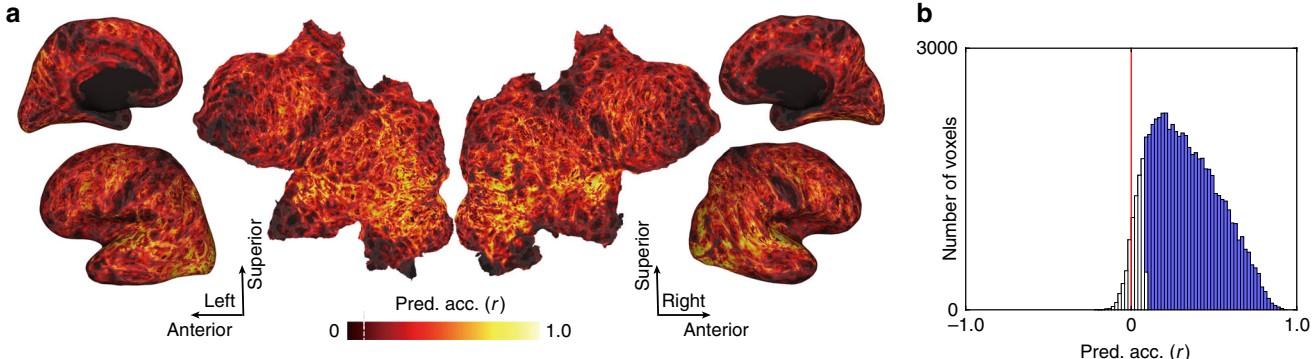

**Fig. 5 Predicting the brain activity of novel tasks using the cognitive factor model. a** Cortical map of model prediction accuracy on inflated and flattened cortical sheets of subject ID01 (Supplementary Fig. 6 shows other subjects). The mean prediction accuracy across the cortex was 0.323 (87.2% of voxels were significant; $p < 0.05$, FDR-corrected; dashed line indicates the threshold). The minimum correlation coefficient for the significance criterion was 0.0846. **b** Histogram of prediction accuracies for all cortical voxels for subject ID01 (Supplementary Fig. 7 shows other subjects). Filled bars indicate voxels that were predicted with significant accuracy.

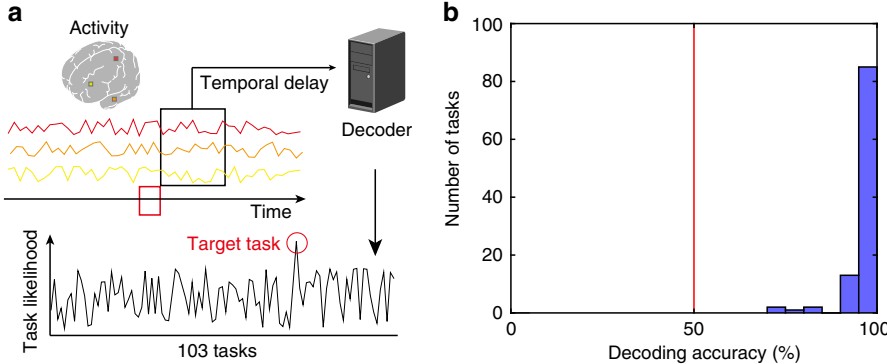

**Fig. 6 Decoding novel tasks using the cognitive factor model. a** A schematic image of the decoding method. Task likelihood was calculated for each target time point using brain activity with hemodynamic temporal delays. The decoding accuracy was examined by a series of binary classifications between the target task (red circle) and the other remaining tasks. **b** Histogram of task decoding accuracies for all tasks for subject ID01 using a binary classification (Supplementary Fig. 9 shows the other subjects). The red line indicates the chance-level accuracy (50%). Filled bars indicate tasks that were decoded with significant accuracy (mean decoding accuracy, 96.8%; all tasks were significant; one-sided sign test, $p < 0.05$, FDR-corrected).

cognitive factors are not inferred in advance. Therefore, in this study, using over 100 naturalistic tasks that broadly sampled the human cognitive space, the prediction accuracy we observed for our model throughout the entire cortex is in clear contrast to the results of previous studies. For example, while our previous modeling attempt using a passive viewing paradigm[4] provided significant predictions for 22% of cortical voxels, which were largely restricted to the occipital and temporal areas, the cognitive factor model in the present study achieved significant predictions for approximately 86% of all cortical voxels. The fact that our model achieved unprecedentedly wide generalizability regarding cortical coverage and multi-task decodability indicates that our task battery represents a sufficient number of samples capable of probing the major proportion of the human cognitive space and provides a baseline for complete characterization of the cognitive space.

By evaluating the similarity between each task cluster or PC and the large reference metadata, we obtained a data-driven interpretation of the task clusters and PCs. Task clusters, which included visual processing tasks, were highly linked with associated cognitive factors, such as "visual" or "perception." In addition, task clusters, including language-related tasks, were linked with "semantic" or "comprehension." These results indicated that metadata-based inference can be useful for obtaining an objective interpretation of task clusters or PCs. Moreover, this method also allowed for the interpretation of task clusters that

may be difficult to label based on the included task types. For example, the task cluster that included "introspection tasks" was found to be related to the "default mode network". Therefore, this metadata-based inference technique demonstrated the applicability of such a data-driven approach for elucidating the brain organization of diverse cognitive functions, without introducing pre-defined assumptions of the task-related cognitive factors.

In the majority of the subjects, task clusters and top PCs contained auditory and visual clusters and components. Thus, it may be argued that the cortical representations of over 100 tasks can be solely explained by the difference of stimulus modality (i.e., sensory input). However, we found that there were task clusters in more abstract cognitive domains, such as memory and language. We also showed that the tasks in the same stimulus domain could be distinguished by decoding analysis. Furthermore, additional analyses with sensorimotor regressors showed significant prediction accuracy covering the whole cerebral cortex. These findings confirmed that these sensory regressors covered the relevant low-level features. Although stimulus modality is an important element in our cognitive space, detailed task representations also included abstract cognitive factors.

Previous studies have identified several clusters and components found in the present study using multiple tasks[15–21]. Compared with these previous works, the current study provides a framework for elucidating more general and quantitative aspects of human cognition in the following points. Using our

broad task sampling paradigm, we revealed a gradual shift in the cognitive space and corresponding cortical organization, from the perceptual to more complex cognitive tasks. HCA and PCA further provided a cortical representational basis for the modeling analyses by demonstrating that the over 100 tasks cover the activity of the whole cerebral cortex and that they were differently organized in each cortical voxel. This is particularly effective in association cortices, such as the IPL, which has been associated with various cognitive domains[22]. By building the relevant encoding and decoding models using data from single subjects, we were able to successfully quantify the generalizability of our model for arbitrary novel task conditions. Such subject-wise modeling may also form the quantitative basis for elucidating the personal traits associated with cognitive functions[23].

Although the tasks used here do not cover the entire domain of human perception and cognition (e.g., they did not cover odor perception, speech, social interaction), our method is applicable to any arbitrary task that can be performed in a scanner. Taken together, our framework provides a powerful step toward the complete modeling of the representations underlying human cognition.

## Methods

**Subjects**. Six healthy subjects (aged 22–33 years, two females; referred to as ID01–ID06) with normal vision and hearing participated in the current experiment. Subjects were all right-handed (laterality quotient = 70–100), as assessed using the Edinburgh inventory[24]. Written informed consent was obtained from all subjects prior to their participation in the study. This experiment was approved by the ethics and safety committee of the National Institute of Information and Communications Technology in Osaka, Japan.

**Stimuli and procedure**. We prepared 103 naturalistic tasks that could be performed without any preexperimental training (see Supplementary Methods for the detailed description of each task; see Supplementary Fig. 10 and Supplementary Note 6 for the behavioral results). Tasks were selected to include as many cognitive domains as possible. Each task had 12 instances; 8 instances were used in the training runs, whereas 4 were used in the test runs. The stimuli were presented on a projector screen inside the scanner (21.0 × 15.8° of visual angle at 30 Hz). The root-mean square of the auditory stimuli was normalized. During scanning, subjects wore MR-compatible ear tips. The experiment was performed for 3 days, with six runs performed each day. Presentation software (Neurobehavioral Systems, Albany, CA, USA) was used to control the stimulus presentation and the collection of behavioral data. To measure button responses, optic response pads with two buttons in each of the left and right hands were used (HHSC-2 × 2, Current Designs, Philadelphia, PA, USA).

The experiment consisted of 18 runs, with 12 training runs and 6 test runs. Each run contained 77–83 trials with duration of 6–12 s per trial. To keep subjects attentive and engaged, and to ensure all runs had the same length, a 2-s feedback for the preceding task (correct or incorrect) was presented 9–13 times per run. In addition to the task, 6 s of imaging without a task was inserted at the beginning and at the end of each run; the former was discarded in the analysis. The duration of a single run was 556 s. In the training runs, task order was pseudorandomized, as some tasks depended on each other and were therefore presented close to each other in time (e.g., the tasks "MemoryDigit" and "MatchDigit"). In the test runs, 103 tasks were presented four times in the same order across all six runs (but with different instances for each repetition). There was no overlap between the instances in the training runs and the test runs. No explanation of the tasks was given to the subjects prior to the experiment. During the fMRI experiment, subjects were instructed on how to perform each task by the instruction text that was shown as a part of the stimuli (see Fig. 1a). Subjects only underwent a short training session on how to use the buttons used to respond.

**MRI data acquisition**. The experiment was conducted using a 3.0 T scanner (TIM Trio; Siemens, Erlangen, Germany) with a 32-channel head coil. We scanned 72 interleaved axial slices that were 2.0-mm thick without a gap, parallel to the anterior and posterior commissure line, using a T2*-weighted gradient-echo multiband echo-planar imaging (MB-EPI) sequence[25] [repetition time (TR) = 2000 ms, echo time (TE) = 30 ms, flip angle (FA) = 62°, field of view (FOV) = 192 × 192 mm², resolution = 2 × 2 mm², MB factor = 3]. We obtained 275 volumes for each run, with each following three dummy images. For anatomical reference, high-resolution T1-weighted images of the whole brain were also acquired from all subjects with a magnetization-prepared rapid acquisition gradient echo sequence (MPRAGE, TR = 2530 ms, TE = 3.26 ms, FA = 9°, FOV = 256 × 256 mm², voxel size = 1 × 1 × 1 mm³).

**fMRI data preprocessing**. Motion correction in each run was performed using the statistical parametric mapping toolbox (SPM8; Wellcome Trust Center for Neuroimaging, London, UK; http://www.fil.ion.ucl.ac.uk/spm/). All volumes were aligned to the first EPI image for each subject. Low-frequency drift was removed using a median filter with a 240-s window. The response for each voxel was then normalized by subtracting the mean response and scaling to unit variance. We used FreeSurfer[26,27] to identify the cortical surfaces from the anatomical data and to register them to the voxels of the functional data. For each subject, the voxels identified in the cerebral cortex were used in the analysis (53,345–66,695 voxels per subject).

**Task-type model**. The task-type model was composed of one-hot vectors, which were assigned 1 or 0 for each time bin, indicating whether one of the 103 tasks was performed in that period. The total number of task-type model features was thus 103.

**Encoding model fitting**. In the encoding model, cortical activity in each voxel was fitted with a finite impulse response model that captured the slow hemodynamic response and its coupling with neural activity[3,28]. The feature matrix $\mathbf{F_E}$ [$T \times 3N$] was modeled by concatenating sets of [$T \times N$] feature matrices with three temporal delays of 2, 4, and 6 s ($T$ = # of samples; $N$ = # of features). The cortical response $\mathbf{R_E}$ [$T \times V$] was then modeled by multiplying the feature matrix $\mathbf{F_E}$ with the weight matrix $\mathbf{W_E}$ [$3N \times V$] ($V$ = # of voxels):

$$\hat{\mathbf{R}}_\mathbf{E} = \mathbf{F_E}\mathbf{W_E}$$

We used an L2-regularized linear regression using the training dataset to obtain the weight matrix $\mathbf{W_E}$. The training dataset consisted of 3336 samples (6672s). The optimal regularization parameter was assessed using 10-fold cross-validation, with the 18 different regularization parameters ranging from 100 to $100 \times 2^{17}$.

The test dataset consisted of 412 samples (824 s, repeated four times). To reshape the data spanning over six test runs into the four times-repeated dataset, we discarded 6 s of the no-task period at the end of each run, as well as the 2-s feedback periods at the end of the third and sixth test runs. Four repetitions of the test dataset were averaged to increase the signal-to-noise ratio. Prediction accuracy was calculated using Pearson's correlation coefficient between the predicted signal and the measured signal in the test dataset. Statistical significance (one-sided) was computed by comparing estimated correlations to the null distribution of correlations between two independent Gaussian random vectors with the same length as the test dataset[3,10]. The statistical threshold was set at $p < 0.05$ and corrected for multiple comparisons using the FDR procedure[29].

**Evaluation of optimal regularization parameters**. To keep the scale of the weight values consistent across subjects, we performed a resampling procedure to assess the optimal regularization parameter used for group HCA and PCA[10]. To this end, for each subject, we randomly divided the training dataset into training samples (80%) and validation samples (20%) and performed model fitting using an L2-regularized linear regression. This procedure was repeated 50 times, with the 18 different regularization parameters ranging from 100 to $100 \times 2^{17}$. The resultant prediction accuracies were averaged across the six subjects for each parameter. We selected the optimal regularization parameter that provided the highest mean prediction accuracy across subjects. This regularization parameter was used for model fitting for group HCA and PCA.

**Hierarchical cluster analysis**. To examine hierarchical relations across tasks, we conducted an HCA. First, we concatenated the weight matrix of predictive voxels of the task-type model across six subjects. Concatenation of the estimated weights was performed to obtain a group-level representation that provides a common basis that is comparable across subjects[4,10]. To choose predictive voxels, for each subject, we selected the voxels that exhibited a significant prediction accuracy, with $p < 0.05$ (with FDR correction, 39,485–56,634 voxels per subject), and averaged three time delays for each task. We then obtained the RSM by calculating the Pearson's correlation coefficients between the averaged weights of all task pairs. A dendrogram of 103 tasks was then described using the task dissimilarity (1—correlation coefficient) as a distance metric, using the minimum distance as a linkage criterion. Each cluster was labeled based on the included cognitive tasks.

To investigate task clusters in a model-independent way, we also conducted HCA using the brain activity of the whole cerebral cortex (Supplementary Fig. 1 and Supplementary Note 1) and visualized the 103 tasks on the two-dimensional space using non-metric multi-dimensional scaling (Supplementary Fig. 3 and Supplementary Note 3).

**Interpretation of cognitive factors related to task clusters**. To interpret the plausible cognitive factors related to the target subclusters obtained in the HCA, we used Neurosynth (http://neurosynth.org; accessed 26 January 2018) as a metadata reference of the past neuroimaging literature[13]. From the approximately 3000 terms in the database, we manually selected 715 terms that covered the comprehensive cognitive factors while also avoiding redundancy. In particular, we removed several plural terms that also had their singular counterpart (e.g.,

"concept" and "concepts") and past tense verbs that also had their present counterpart (e.g., "judge" and "judged") from the dataset. We also excluded those terms that indicated anatomical regions (e.g., "parietal"). We used the reverse-inference map of the Neurosynth database for each of the 715 selected terms. The reverse-inference map indicated the likelihood of a given term being used in a study if the activity was observed at a particular voxel. Each reverse-inference map in the MNI152 space was then registered to the subjects' reference EPI data using FreeSurfer[26,27]. For each of the cortical maps of the task cluster weight matrix (Supplementary Fig. 2 and Supplementary Note 2), we calculated Pearson's correlation coefficients with the 715 registered reverse-inference maps, which resulted in a cognitive factor vector with 715 elements. Terms with higher correlation coefficient values were regarded as contributing more to the target cluster.

**Principal component analysis of task-type weights.** For each subject, we performed PCA on the weight matrix of the task-type model concatenated across six subjects. We selected the voxels that showed significant prediction accuracy with $p < 0.05$ (with FDR correction, 39,485–56,634 voxels per subject) and averaged three time delays for each task. To show the structure of the cognitive space, 103 tasks were mapped onto the two-dimensional space using the loadings of PC1 (1st PC) and PC2 as the x-axis and y-axis, respectively. The tasks were further colored in red, green, and blue, based on the relative PCA loadings in PC1, PC2, and PC3, respectively. To obtain an objective interpretation of the PCs, we also performed metadata-based inference of the cognitive factors related to each PC (Supplementary Table 3 and Supplementary Note 4).

To represent the cortical organization of the cognitive space for each subject, we extracted and normalized the PCA scores from each subject's voxels. The resultant cortical map indicated the relative contribution of each cortical voxel to the target PC (denoted as 'PCA score map', Supplementary Fig. 4 and Supplementary Note 5). By combining the PCA score maps from the top three PCs for each subject, we were able to visualize how each cortical voxel is represented by the associated cognitive clusters. Each cortical voxel was colored based on the relative PCA scores of PC1, PC2, and PC3, corresponding to the color of the tasks in the two-dimensional space.

**Hierarchical model.** To further quantify the importance of the estimated hierarchy, we constructed a hierarchical model based on the result of HCA. We first conducted HCA using data from five subjects (excluding the target subject's data). This dendrogram (cluster tree) includes 102 non-terminal nodes (the terminal nodes correspond to the 103 tasks). The feature matrix of the hierarchical model consisted of 102-element binary vectors, which were assigned either 1 or 0 for each time bin, indicating whether any tasks subordinated by the target non-terminal node were performed in that period.

**Cognitive factor model.** To obtain task representations using continuous features in the human cognitive space, we transformed sparse task-type features into the latent cognitive factor feature space (Fig. 1d). We used the reverse-inference map of the Neurosynth database[13] for each of the 715 terms selected. Each reverse-inference map in the Neurosynth database in MNI152 space was registered to the subjects' reference EPI data using FreeSurfer[26,27].

We then calculated the correlation coefficients between the weight map for each task in the task-type model and the registered reverse-inference maps. This resulted in the [103 × 715] coefficient matrix. We next obtained the CTF for each subject by averaging the coefficient matrices of the other five subjects. The CTF served to transform the feature values of the 103 tasks into the 715-dimensional latent feature space. The feature matrix of the cognitive factor model was then obtained by multiplying the CTF with the feature matrix of the task-type model. Note that the CTF (and the resultant feature matrix) of each target subject was independent of their own data. The total number of cognitive factor model features was 715.

**Encoding model fitting with sensorimotor regressors.** To evaluate the possible effect of low-level sensorimotor features on model predictions, we performed an additional encoding model fitting while regressing out the sensorimotor components. To this end, we concatenated motion energy (ME) model features (visual), modulation transfer function (MTF) model features (auditory), and button response (BR) model features (motor) with the original feature matrix during model training (see the following subsections for details). ME model features were obtained by applying three-dimensional spatiotemporal Gabor wavelet filters to the visual stimuli[3]. MTF model features were obtained by applying spectro-temporal modulation-selective filters to the cochleogram of the auditory stimuli[30]. BR model features were obtained based on the number of button responses made by each subject. Model testing excluded the sensorimotor regressors from the concatenated feature matrix and the corresponding weight matrix. This analysis revealed that the model prediction accuracy was independent of low-level sensorimotor features.

**Motion energy model.** We used the ME model that has been used in previous studies[3,31,32] and provided in a public repository (https://github.com/gallantlab/motion_energy_matlab). First, movie frames and pictures were spatially downsampled to 96 × 96 pixels. The RGB pixel values were then converted into the Commission International de l'Eclairage (CIE) LAB color space, and the color

information was subsequently discarded. The luminance (L∗) pattern was passed through a bank of three-dimensional spatiotemporal Gabor wavelet filters. The outputs of the two filters with orthogonal phases (quadrature pairs) were squared and summed to yield local ME. ME was compressed with a log-transformation and temporally downsampled to 0.5 Hz. Filters were tuned to six spatial frequencies (0, 1.5, 3.0, 6.0, 12.0, 24.0 cycles per image) and three temporal frequencies (0, 4.0, 8.0 Hz), without directional parameters. Filters were positioned on a square grid that covered the screen. The adjacent filters were separated by 3.5 standard deviations of their spatial Gaussian envelopes. The total number of ME model features was 1395.

**Modulation transfer function model.** A sound cochleogram was generated using a bank of 128 overlapping bandpass filters ranging from 20 to 10,000 Hz[33]. The window size was set to 25 ms and the hop size to 10 ms. The filter output was averaged across 2 s (TR). We further extracted the features from the MTF model[30] which we provided in a public repository (https://osf.io/ea2jc/). For each cochleogram, a convolution with modulation-selective filters was then calculated. The outputs of the two filters with orthogonal phases (quadrature pairs) were squared and summed to yield the local modulation energy[3]. Modulation energy was then log-transformed, averaged across 2 s, and further averaged within each of the 10 nonoverlapping frequency ranges logarithmically spaced along the frequency axis. The filter outputs of the upward and downward sweep directions were used. Modulation-selective filters were tuned to five spectral modulation scales (0.50, 1.0, 2.0, 4.0, 8.0 cycles per octave) and five temporal modulation rates (4.0, 8.0, 16.0, 32.0, 64.0 Hz). The total number of MTF model features was 1000.

**Button response model.** The BR model was constructed based on the number of button responses within 1 s for each of the four buttons, with the right two buttons pressed by the right thumb and the left two buttons pressed by the left thumb. The total number of BR model features was four.

**Decoding model fitting.** In the decoding model, the cortical response matrix $\mathbf{R_D}$ [$T \times 3\,V$] was modeled using concatenating sets of [$T \times V$] matrices with temporal delays of 2, 4, and 6 s. The feature matrix $\mathbf{F_D}$ [$T \times N$] was modeled by multiplying the cortical response matrix $\mathbf{R_D}$ with the weight matrix $\mathbf{W_D}$ [$3\,V \times N$]:

$$\hat{\mathbf{F}}_\mathbf{D} = \mathbf{R_D}\mathbf{W_D}$$

The weight matrix $\mathbf{W_D}$ was estimated using an L2-regularized linear regression with the training dataset, following the same procedure for the encoding model fitting.

To test decoding performance in a model-independent way, we also decoded over 100 tasks directly from brain activity using a support vector machine (Supplementary Fig. 11 and Supplementary Note 7).

**Encoding and decoding with novel tasks.** In order to examine the generalizability of our models, we performed encoding and decoding analyses with novel tasks not used during model training (Fig. 1e). We randomly divided the 103 tasks into five task groups. A single task group contained 20–21 tasks. We performed five independent model fittings, each with a different task group as the target. From the training dataset, we excluded the time points during which the target tasks were performed and those within 6 s after the presentation of the target tasks. In the test dataset, we used only the time points during which the target tasks were performed and those within 6 s after the presentation of the target tasks. This setting allowed us to assume that the activity induced by the target task group and that induced by the other four task groups (training task groups) did not overlap, thus enabling us to investigate prediction and decoding accuracies for novel tasks. We performed encoding and decoding model fitting with the training task group, which was composed of 82–83 tasks. For model testing, we concatenated the predicted responses or decoded features of the five task groups. Responses or features for the time points that were duplicated were then averaged across the five task groups. Note that encoding and decoding with the novel tasks were only possible with the cognitive factor model, because the original tasks needed to be transformed into the latent feature space.

In a further analysis, we used a random shuffling procedure to obtain a null distribution of mean prediction accuracy (Supplementary Fig. 12). Specifically, all elements of the [715 × 412] feature matrix of the cognitive factor encoding model in the test dataset were randomly shuffled in an element-wise manner, and predicted responses were calculated by multiplying the shuffled feature matrix with the original weight matrix. The mean prediction accuracy across all voxels was calculated. This procedure was repeated 1000 times.

For the decoding analysis with novel tasks, we measured the similarity between the CTF of each task and each decoded cognitive factor vector using Pearson's correlation coefficients for each time point. We refer to the correlation coefficient as the 'task likelihood'. We then calculated the time-averaged task likelihoods for each task using the one-vs.-one method. For each target task, a series of binary classifications was performed between the target task and each of the remaining 102 tasks. The decoding accuracy was then calculated as a percentage that the target task had higher task likelihood in this procedure. The statistical significance of the decoding accuracy was tested for each task using the one-sided sign test ($p <$

0.05, with FDR correction). See Supplementary Note 8 for the comparison between prediction and decoding results.

Additionally, we used a random shuffling procedure to obtain a null distribution of mean decoding accuracy (Supplementary Fig. 13). Specifically, all elements of the [103 × 715] CTF matrix were randomly shuffled in an element-wise manner, and the task likelihood was measured using Pearson's correlation coefficient between the shuffled CTF of each task and each decoded cognitive factor vector. The mean decoding accuracy across all tasks was calculated. This procedure was repeated 1000 times.

All model fitting and analyses were conducted using custom software written on MATLAB. For data visualization on the cortical maps, pycortex was used[34].

**Reporting summary**. Further information on research design is available in the Nature Research Reporting Summary linked to this article.

## Data availability
The source data underlying all Figures, Supplementary Figures, and Tables except Fig. 1 are provided as a Source Data file at the Open Science Framework (OSF, https://osf.io/ea2jc/). The raw MRI data are available at the OpenNeuro.org (https://openneuro.org/datasets/ds002306).

## Code availability
The MATLAB code used in the current study and the datasets generated and/or analyzed during the current study are available at the OSF repository (https://osf.io/ea2jc/).

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

## Acknowledgements
We thank MEXT/JSPS KAKENHI (grant numbers JP17K13083 and JP18H05091 in #4903 (Evolinguistics) for T.N., JP15H05311 and JP18H05522 for S.N.) as well as JST CREST JPMJCR18A5 and ERATO JPMJER1801 (for S.N.) for the partial financial support of this study. The funders had no role in the study design, data collection, and analysis, decision to publish, or preparation of the manuscript.

## Author contributions
T.N. and S.N. designed the study; T.N. collected and analyzed the data; T.N. and S.N. wrote the manuscript.

## Competing interests
The authors declare no competing interests.
