## [Peer Review File · Nature Communications]

Reviewers' Comments:

Reviewer #1:

Remarks to the Author:

This paper reports an analysis of a unique dataset in which six individuals performed a large number of tasks, and the data were analyzed using encoding models based on the cognitive features of the tasks. This work is a logical extension of previous work that has used passive audiovisual tasks, and extends that earlier work significantly. In particular, the demonstration that patterns of brain activity can be predicted for new cognitive tasks is novel and important.

Despite my enthusiasm, I also have some concerns. The first is that much of the paper is simply descriptive, providing an overview of low-dimensional structure obtained through clustering and dimensionality reduction. There is nothing particularly novel about these results, and in fact the presentation of them seems less compelling than many previous explications of low-dimensional structure in relation to cognitive functions (e.g. by Yeo et al.).

My second concern regards the decoding accuracy: The results are just too good. Whenever I have seen decoding results for any kind of complex task approaching 100%, it has always turned out to be an error that allows test set information to creep into the training set. I am worried that something like this could have happened here. A complication is that the analysis stream is deeply complex, so it's very difficult for me to see exactly where this might have happened - and I fear that looking at the code would not really help, given the complexity of the analysis procedure. I realize that this is a frustrating comment, as there is littler that the authors could say that could directly contradict it. However, it would certainly be helpful to obtain null distributions by shuffling the cognitive function matrix - though this needs to be done for the entire processing stream.

Third, I generally found the analyses unnecessarily complicated and non-standard. Here are a few of my concerns:

- * It seems odd to concatenate across subjects, and it wasn't always clear what was being concatenated.
- * it is unclear how subjects actually knew how to perform the tasks, given that "No explanation of the tasks was given to the subjects prior to the experiment."
- * the motivation is not clear for fitting three delayed regressors (and then averaging them later) as opposed to the standard approach in which a single regressor is convolved with a canonical hemodynamic response
- * "Prediction accuracy was calculated using Pearson's correlation coefficient between the predicted signal and the measured signal in the test dataset." - The absolute error should also be reported, as should null distributions obtained by shuffling the design matrix.
- * The method used to identify optimal hyperparameters is referred to as "bootstrapping" but it doesn't actually appear to be a true bootstrap (which involves sampling with replacement). rather it seems to be a resampling method and should be correctly labeled as such.
- * It is not clear how the dimensionality of the clustering tree cut (of 6) was determined.

Finally, the data analysis methods are missing crucial details - please consult the OHBM COBIDAS guidelines and report all required details. I would also strongly encourage the authors to share the data and code openly, rather than upon request, as this is becoming standard in our field.

Reviewer #2:

Remarks to the Author:

Encoding models of fMRI during 103 cognitive tasks: pushing the envelope of human cognitive brain imaging
[18R7]

Nakai and Nishimoto (pp2019) had each of six subjects perform 103 naturalistic cognitive tasks during functional magnetic resonance imaging (fMRI) of their brain activity. This type of data could eventually enable us to more compellingly characterize the localization of cognitive task components across the human brain.

What is unique about this paper is the fact it explores the space of cognitive tasks more systematically and comprehensively than any previous fMRI study I am aware of. It's important to have data from many tasks in the same subjects to more quantitatively model how cognitive components, implemented in different parts of the brain, contribute in combination to different tasks.

The authors describe the space of tasks using a binary task-type model (with indicators for task components) and a continuous cognitive-factor model (with prior information from the literature incorporated via Neurosynth). They perform encoding and decoding analyses and investigate the clustering of task-related brain activity patterns. The model-based analyses are interesting, but also a bit hard to interpret, because they reveal the data only indirectly: through the lens of the models – and the models are very complex. It would be good to see some more basic "data-driven" analyses, as the title suggests.

However, the more important point is that this is a visionary contribution from an experimental point of view. The study pushes the envelope of cognitive fMRI. The biggest novel contributions are:

the task set (with its descriptive models)

the data (in six subjects)

Should the authors choose to continue to work on this, my main suggestions are (1) to add some more interpretable data-driven analyses, and (2) to strengthen the open science component of the study, so that it can form a seed for much future work that builds on these tasks, expanding the models, the data, and the analyses beyond what can be achieved by a single lab.

This rich set of tasks and human fMRI responses deserves to be analyzed with a wider array of models and methods in future studies. For example, it would be great in the future to test a wide variety of task-descriptive models. Eventually it might also be possible to build neural network models that can perform the entire set of tasks. Explaining the measured brain-activity with such brain-computational models would get us closer to understanding the underlying information processing. In addition, the experiment deserves to be expanded to more subjects (perhaps 100). This could produce a canonical basis for revisiting human cognitive fMRI at a greater level of rigor. These directions may not be realistic for a single study or a single lab. However, this paper could be seminal to the pursuit of these directions as an open science endeavor across labs.

Improvements to consider if the authors chose to revise the paper

(1) Reconsider the phrase "data-driven models" (title)

The phrase "data-driven models" suggests that the analysis is both data-driven and model-based. This suggests the conceptualization of data-driven and model-based as two independent dimensions.

In this conceptualization, an analysis could be low on both dimensions, restricting the data to a small set (e.g. a single brain region) and failing to bring theory into the analysis through a model of some complexity (e.g. instead computing overall activation in the brain region for each experimental condition). Being high on both dimensions, then, appears desirable. It would mean that the assumptions (though perhaps strong) are explicit in the model (and ideally justified), and that the data still richly inform the results.

Arguably this is the case here. The models the authors used have many parameters and so the data richly inform the results. However, the models also strongly constrain the results (and indeed changing the model might substantially alter the results - more on that below).

But an alternative conceptualization, which seems to me more consistent with popular usage of these terms, is that there is a tradeoff between data-driven and model-based. In this conceptualization the overall richness of the results (how many independent quantities are reported) is considered a separate dimension. Any analysis combines data and assumptions (with the latter ideally made explicit in a model). If the model assumptions are weak (compared to the typical study in the same field), an analysis is referred to as data-driven. If the model assumptions are strong, then an analysis is referred to as model-driven. In this conceptualization, "data-driven model" is an oxymoron.

(2) Perform a data-driven (and model-independent) analysis of how tasks are related in terms of the brain regions they involve

"A sparse task-type encoding model revealed a hierarchical organization of cognitive tasks, their representation in cognitive space, and their mapping onto the cortex." (abstract)

I am struggling to understand (1) what exact claims are made here, (2) how they are justified by the results, and (3) how they would constrain brain theory if true. The phrases "organization of cognitive tasks" and "representation in cognitive space" are very vague.

The term hierarchical (together with the fact that a hierarchical cluster analysis was performed) suggests that (a) the activity patterns fall in clusters rather than spreading over a continuum and (b) the main clusters contain nested subclusters.

However, the analysis does not assess the degree to which the task-related brain activity patterns cluster. Instead a complex task-type model (whose details and influence on the results the reader cannot assess) is interposed. The model filters the data (for example preventing unmodeled task components from influencing the clustering). The outcome of clustering will also be affected by the prior over model weights.

A simpler, more data-driven, and interpretable analysis would be to estimate a brain activity pattern for each task and investigate the representational geometry of those patterns directly. It would be good to see the representational dissimilarity matrix and/or and visualization (MDS or t-SNE) of these patterns.

To formally address whether the patterns fall into clusters (and hierarchical clusters), it would be ideal to inferentially compare cluster (and hierarchical cluster) models to continuous models. For example, one could fit each model to a training set and assess whether the models' predictive performance differs on an independent test set. (This is in contrast to hierarchical cluster analysis, which assumes a hierarchical cluster structure rather than inferring the presence of such a structure from the data.)

(3) Perform a simple pairwise task decoding analysis

It's great that the decoding analysis generalizes to new tasks. But this requires model-based generalization. It would be useful, additionally, to use decoding to assess the discriminability of the task-related activity patterns in a less model-dependent way.

One could fit a linear discriminant for each pair of tasks and test on independent data from the same subject performing the same two tasks again. (If the accuracy were replaced by the linear discriminant t value or crossnobis estimator, then this could also form the basis for point (2) above.)

"A cognitive factor encoding model utilizing continuous intermediate features by using metadata-based inferences predicted brain activation patterns for more than 80 % of the cerebral cortex and decoded more than 95 % of tasks, even under novel task conditions." (abstract)

The numbers 80% and 95% are not meaningful in the absence of additional information (more than 80% of the voxel responses predicted significantly above chance level, and more than 95% of the tasks were significantly distinct from at least some other tasks). You could either add the information needed to interpret these numbers to the abstract or remove the numbers from the abstract. (The abstract should be interpretable in isolation.)

--Nikolaus Kriegeskorte

Reviewer #3:

Remarks to the Author:

Nakai and Nishimoto collect BOLD responses while six participants perform over a hundred varied tasks, from perceptual to motor to emotional to linguistic, etc. They then analyze the whole-brain data with a pair of voxelwise encoding models in an effort to characterize brain responses to this diverse array of tasks. They find that a continuous latent space is able to account for a reasonable amount of the variance of responses across nearly the entire brain and that such a latent space is able to generalize to novel tasks, as well.

This paper is largely methodological—I'm not totally sure what scientific inference we can glean from this paper. With that being said, I find this paper interesting and innovative, a creative extension of the voxelwise encoding approach to whole-brain task-driven responses. I think it would be a solid and exciting contribution to the literature.

Major comments:

I frankly don't have any major criticisms. The only thing I wonder is if there is something beyond methodology that can be taken away from this paper, and if there is any such takeaway it would be great for the authors to communicate it.

Minor comments:

I found myself having to go to the methods/supplement repeatedly for items that I could imagine would be in the main text/figures; e.g.,

1. Please add a list of many of the tasks to figure 1; all 100 may be too much, but surely 10-20 could fit in a new panel

2. In the results section it would be useful to have listed the number of subjects and amount of scanning data per subject
3. The description of the cognitive factor feature space from Neurosynth in the main text is lacking (e.g., lines 191-195). A reader should be able to understand what you did from that portion alone, and the text there could be improved to clarify what actually occurred.

We are grateful to the reviewers for reviewing our manuscript and providing their valuable comments. We have modified our manuscript based on these comments and have provided our responses to their comments below. For clarity, our responses are given after each of the reviewers' comments. Passages that have been newly inserted into the manuscript have been underlined.

Reviewer #1 (Remarks to the Author):

Comment 1: Despite my enthusiasm, I also have some concerns. The first is that much of the paper is simply descriptive, providing an overview of low-dimensional structure obtained through clustering and dimensionality reduction. There is nothing particularly novel about these results, and in fact the presentation of them seems less compelling than many previous explications of low-dimensional structure in relation to cognitive functions (e.g. by Yeo et al.).

Answer: The main novelty of this study lies in the encoding and decoding modelling of novel tasks using the cognitive factor model, as described in Figures 5 and 6. We were successfully able to predict and decode the tasks that were not used in model training by representing all tasks in the latent feature space of the cognitive factor model.

We agree with the reviewer that Figures 2– 4 demonstrate data in a descriptive manner. However, we believe that these figures provide important information about how the 103 tasks were related to each other as well as how they are spatially organised across the whole cerebral cortex for each participant. Without this information, we might not know the representational basis of the estimated models. To clarify this point, we added the following text to the Introduction section of the manuscript:

HCA and PCA further provided a cortical representational basis for the modeling analyses by demonstrating that the over 100 tasks cover the activity of the whole cerebral cortex and that they were differently organized in each cortical voxel.

Furthermore, to examine the importance of the estimated hierarchical structure (refer to the response to reviewer 2's comment 4), we have performed an additional encoding modelling analysis that reflects the hierarchical relationship. We found that the model that used clustering information outperformed the original task-type model without hierarchy-based information. This result provides further quantitative support to the finding that estimated hierarchy captures the functional structure of the cortex.

We have also added the suggested literature in the Discussion section and citation to the References section:

'Previous studies have identified several clusters and components found in the present study using multiple tasks¹⁵⁻²¹'

21. Thomas Yeo, B. T. et al. Functional specialization and flexibility in human association cortex. Cereb. Cortex 25, 3654–3672 (2015).

Comment 2: My second concern regards the decoding accuracy: The results are just too good. Whenever I have seen decoding results for any kind of complex task approaching 100%, it has always turned out to be an error that allows test set information to creep into the training set. I am worried that something like this could have happened here. A complication is that the analysis stream is deeply complex, so it's very difficult for me to see exactly where this might have happened - and I fear that looking at the code would not really help, given the complexity of the analysis procedure. I realize that this is a frustrating comment, as there is littler that the authors could say that could directly contradict it. However, it would certainly be helpful to obtain null distributions by shuffling the cognitive function matrix - though this needs to be done for the entire processing stream.

Answer: We understand the reviewer's concern. We have rechecked our scripts and have not found any data leakage between the training and test datasets that were measured independently. However, thanks to the reviewer, we noticed minor mistakes in the decoding script and modified them, leading to slight changes in the decoding accuracy (described at the end of this Response letter). Note that several groups, including ours, have shown that multivariate visual decoding models can achieve high accuracy (>90%) even for bigger set sizes (e.g. Kay et al., 2008 Nature; Nishimoto et al., 2011 Current Biology). Our current study provides an expansion of this work using these models from vision to higher-order cognition. We share the enthusiasm of other authors that a modelling approach can achieve this level of accuracy.

In accordance with the reviewer's suggestion and to further test the reliability of our results, we shuffled the cognitive transform functions during decoding analysis 1,000 times and obtained a null distribution of mean decoding accuracy across the 103 tasks. We found that the distribution is mainly centred around the chance level (50%) for all subjects and that the decoding accuracy obtained in the current study (mean \pm SD, $96.5 \pm 0.9\%$) is far greater than that obtained by chance. We have shown this shuffling analysis in Supplementary Figure 12 and have added the following text to the Methods section of the manuscript:

Additionally, we used a random shuffling procedure to obtain a null distribution of mean decoding accuracy (Supplementary Figure 13). During the decoding analysis, CTF was randomly shuffled, and the mean decoding accuracy across all tasks was calculated. This procedure was repeated 1,000 times.

In the caption for Supplementary Figure 13, we have added the following text:

Supplementary Figure 13. Distribution of decoding accuracy with random shuffling. The histogram of mean decoding accuracy generated by a random shuffling procedure is plotted for subjects ID01–ID06.

Comment 3: Third, I generally found the analyses unnecessarily complicated and non-standard. Here are a few of my concerns:

* It seems odd to concatenate across subjects, and it wasn't always clear what was being concatenated.

Answer: We concatenated the estimated weights across subjects in order to obtain a group-level representation (e.g. Huth et al., 2012 Neuron; Huth et al., 2016 Nature). Without concatenation, the PCs and clusters would be different for each subject. This would make the comparison and interpretation of the representation across subjects harder. To clarify the above point, we added the following text to the Methods section:

Concatenation of the estimated weights was performed in order to obtain a group-level representation that provides a common basis that is comparable across subjects^{4,10}.

Comment 4: it is unclear how subjects actually knew how to perform the tasks, given that "No explanation of the tasks was given to the subjects prior to the experiment."

Answer: We designed our tasks to be simple and natural to perform. The subjects could understand how to perform each task by looking at the instruction text that appeared with it. For the example, as shown in Figure 1b, the instruction for the 'EmotionVoice' task was indicated by the phrase 'Sad voice?', which was presented as part of the stimulus. To clarify this, we have added the following text to the Methods section:

During the fMRI experiment, subjects were instructed on how to perform each task by the instruction text that was shown as a part of the stimuli (see Fig. 1a).

Comment 5: the motivation is not clear for fitting three delayed regressors (and then averaging them later) as opposed to the standard approach in which a single regressor is convolved with a canonical hemodynamic response

Answer: We used a finite impulse response (FIR) model (Kay et al., Human Brain mapping 2008), which has also been used in previous studies by Nishimoto et al. (Curr. Biol. 2011), Huth et al. (Neuron, 2012), Huth et al. (Nature, 2016), and so on. A canonical hemodynamic response function (HRF) depends on an assumption about its shape and is likely to be suboptimal for modelling whole-brain voxel-wise activity patterns where HRF shapes vary across different subjects and brain regions (Handwerker et al., 2004 NeuroImage). We therefore modified the following text in the Methods section:

‘cortical activity in each voxel was fitted with a finite impulse response model that captured the slow hemodynamic response and its coupling with neural activity^{3,27},

We have added the following citation to the References section:

27. Kay, K. N., David, S. V., Prenger, R. J., Hansen, K. A. & Gallant, J. L. Modeling low-frequency fluctuation and hemodynamic response timecourse in event-related fMRI. Hum. Brain Mapp. 29, 142–156 (2008).

Comment 6: "Prediction accuracy was calculated using Pearson's correlation coefficient between the predicted signal and the measured signal in the test dataset. - The absolute error should also be reported, as should null distributions obtained by shuffling the design matrix.

Answer: Due to the nature of BOLD signals that require pre-normalisations, only the relative signal of the evoked time course is functionally interpretable and it is difficult to calculate the absolute error in a meaningful way. We therefore considered the Pearson's correlation coefficient to be appropriate because it is a scale-free measure.

With reference to the reviewer's suggestion and to obtain null distributions, we shuffled the feature matrix (equivalent to the design matrix) of the cognitive factor model 1,000 times and obtained a null distribution of prediction accuracy across all cortical voxels. We found that the distribution is centred around the zero correlation ($r = 0$) for all subjects and thus significantly deviates from the prediction accuracy obtained in the current study (mean \pm SD, 0.322 ± 0.042).

We have shown this result by adding the distribution of shuffled prediction accuracies to Supplementary Figure 5. For the purpose of comparison, we added the distribution of prediction accuracy for the original cognitive factor model (under novel task condition) to Figure 5b and Supplementary Figure 5.

We also added the following text to the Methods section:

In a further analysis, we used a random shuffling procedure to obtain a null distribution of mean prediction accuracy (Supplementary Fig. 12). Specifically, the feature matrix of the cognitive factor encoding model was randomly shuffled, and the mean prediction accuracy across all voxels was calculated. This procedure was repeated 1,000 times.

We have added the following text to the caption of Supplementary Figures 5 and 12:

Supplementary Figure 5. Distribution of prediction accuracy using the cognitive factor model under novel task conditions. Histogram of prediction accuracies for all cortical voxels for subjects ID02-ID06. Filled bars indicate voxels that were predicted with significant accuracy.

Supplementary Figure 12. Distribution of prediction accuracy with random shuffling. The histogram of mean prediction accuracy generated by a random shuffling procedure is plotted for subjects ID01–ID06.

Comment 7: The method used to identify optimal hyperparameters is referred to as bootstrapping; but it doesn't actually appear to be a true bootstrap (which involves sampling with replacement). rather it seems to be a resampling method and should be correctly labeled as such.

We thank the reviewer for catching this. We have modified this point in Line 385 to 'resampling procedure'.

Comment 8: It is not clear how the dimensionality of the clustering tree cut (of 6) was determined.

Answer: This was determined by visual inspection for a descriptive purpose. However, the number of largest clusters (of 6) is not crucial to this study. We can apply the same analysis (e.g. reverse inference using Neurosynth) for any of the subclusters in the dendrogram and the number of clusters does not affect the other analyses (such as the decoding analysis).

To clarify this point, we have added the following text to the Results section:

Although six clusters were determined by visual inspection for a descriptive purpose, the same analyses can be performed on any subclusters in the dendrogram.

Comment 9: Finally, the data analysis methods are missing crucial details - please consult the OHBM COBIDAS guidelines and report all required details.

Answer: We checked the OHBM COBIDAS guidelines and noticed that several pieces of information were missing from the Methods section. We added the following pieces of text to the Methods section:

(SPM8; Wellcome Trust Centre for Neuroimaging, London, UK; <http://www.fil.ion.ucl.ac.uk/spm/>)

Presentation software (Neurobehavioral Systems, Albany, CA, USA) was used to control the stimulus presentation and the collection of behavioral data.

To measure button responses, optic response pads with two buttons in each of the left and right hands were used (HHSC-2x2, Current Designs, Philadelphia, PA, USA).

All model fitting and analyses were conducted using custom software written on MATLAB. For data visualization on the cortical maps, pycortex was used³¹.

Comment 10: I would also strongly encourage the authors to share the data and code openly, rather than upon request, as this is becoming standard in our field.

Answer: We are working toward making our MRI data and code openly available, although further administrative arrangements are necessary, due to the current data policy held by our institution.

Reviewer #2 (Remarks to the Author):

Comment 1: Should the authors choose to continue to work on this, my main suggestions are (1) to add some more interpretable data-driven analyses, and (2) to strengthen the open science component of the

study, so that it can form a seed for much future work that builds on these tasks, expanding the models, the data, and the analyses beyond what can be achieved by a single lab.

Answer: We thank the reviewer for these constructive suggestions. We fully agree and (1) have added more data-driven analyses (see below) and (2) will make the data and models openly available for future work.

Comment 2: Improvements to consider if the authors chose to revise the paper

(1) Reconsider the phrase "data-driven models" (title)

The phrase "data-driven models" suggests that the analysis is both data-driven and model-based. This suggests the conceptualization of data-driven and model-based as two independent dimensions.

In this conceptualization, an analysis could be low on both dimensions, restricting the data to a small set (e.g. a single brain region) and failing to bring theory into the analysis through a model of some complexity (e.g. instead computing overall activity in the brain region for each experimental condition). Being high on both dimensions, then, appears desirable. It would mean that the assumptions (though perhaps strong) are explicit in the model (and ideally justified), and that the data still richly inform the results.

Arguably this is the case here. The models the authors used have many parameters and so the data richly inform the results. However, the models also strongly constrain the results (and indeed changing the model might substantially alter the results - more on that below).

But an alternative conceptualization, which seems to me more consistent with popular usage of these terms, is that there is a tradeoff between data-driven and model-based. In this conceptualization the overall richness of the results (how many independent quantities are reported) is considered a separate dimension. Any analysis combines data and assumptions (with the latter ideally made explicit in a model). If the model assumptions are weak (compared to the typical study in the same field), an analysis is referred to as data-driven. If the model assumptions are strong, then an analysis is referred to as model-driven. In this conceptualization, "data-driven model" is an oxymoron.

Answer: We agree with the reviewer and have modified the title as follows:

‘Quantitative models reveal the organization of diverse cognitive functions in the brain’

We have also modified the related phrases in the Introduction section:

‘we combined ~~data-driven~~ encoding modelling and metadata-based reverse inference’

Comment 3: (2) Perform a data-driven (and model-independent) analysis of how tasks are related in terms of the brain regions they involve

"A sparse task-type encoding model revealed a hierarchical organization of cognitive tasks, their representation in cognitive space, and their mapping onto the cortex." (abstract)

I am struggling to understand (1) what exact claims are made here, (2) how they are justified by the results, and (3) how they would constrain brain theory if true. The phrases "organization of cognitive tasks" and "representation is cognitive space" are very vague.

The term hierarchical (together with the fact that a hierarchical cluster analysis was performed) suggests that (a) the activity patterns fall in clusters rather than spreading over a continuum and (b) the main clusters contain nested subclusters.

However, the analysis does not assess the degree to which the task-related brain activity patterns cluster. Instead a complex task-type model (whose details and influence on the results the reader cannot assess) is interposed. The model filters the data (for example preventing unmodeled task components from influencing the clustering). The outcome of clustering will also be affected by the prior over model weights.

A simpler, more data-driven, and interpretable analysis would be to estimate a brain activity pattern for each task and investigate the representational geometry of those patterns directly. It would be good to see the representational dissimilarity matrix and/or and visualization (MDS or t-SNE) of these patterns.

Answer: We agree with the reviewer that the model filters the data. Therefore, we performed hierarchical cluster analysis (HCA) using brain activity (of the whole cerebral cortex) in the training dataset without performing encoding modelling (Supplementary Figure 9). We found that there were very few changes from the original HCA following the new analysis. The task-type encoding model and the components of the largest clusters (visual, auditory, motor, memory, language, and introspection) were unchanged. Accordingly, we added the following text to the Methods section:

To investigate task clusters in a model-independent way, we also conducted HCA using brain activity of the whole cerebral cortex (Supplementary Fig. 9) and visualized the 103 tasks on the two-dimensional space using non-metric multi-dimensional scaling (Supplementary Fig. 10).

We added the following text to the Results section:

We also obtained a similar hierarchy pattern using an RSM calculated directly from brain activity (see Supplementary Figure 9).

The following pieces of text were added to the Supplementary information:

Supplementary Figure 9. Hierarchical cluster analysis using brain activity. a, Representational similarity matrix of the 103 tasks, reordered according to the hierarchical cluster analysis (HCA) using brain activity of the whole cerebral cortex (concatenated across subjects). The dendrogram shown in the top panel represents the results of the HCA. The six largest clusters were named after the included task types.

HCA using brain activity

To investigate task clusters in a model-independent way, we conducted HCA using brain activity of the whole cerebral cortex, concatenated across six subjects. Brain activity during the task period with three time delays was averaged for each task. The RSM and dendrogram were obtained following the same procedure as the HCA using a task-type weight matrix. Each cluster was labelled according to the cognitive tasks included. The resultant RSM and dendrogram were very similar to those obtained using the task-type weight matrix (Fig. 3).

We have also tested MDS based on the dissimilarity matrix obtained using brain activity (Supplementary Figure 10). The result was very similar to the PCA visualisation (Fig. 4). Hence, we have added the following text to the Results section:

We also obtained a similar distribution using a multi-dimensional scaling directly applied to brain activity (Supplementary Figure 10).

We have added the following pieces of text to the Supplementary information:

Supplementary Figure 10. Visualization of cognitive space using multi-dimensional scaling. A total of 103 tasks were mapped onto the two-dimensional cognitive space based on the 1st and 2nd dimensions obtained using a non-metric multi-dimensional scaling. Colors indicate the six largest clusters depicted in Supplementary Fig. 9. All task names are labeled in white.

Visualization of over 100 tasks using multi-dimensional scaling

To further investigate the representational relationship among the 103 tasks in a model-independent way, we conducted a non-metric multi-dimensional scaling analysis and visualized 103 tasks on the two-dimensional space based on the 1st and 2nd dimensions (Supplementary Fig. 10). We found that tasks in the same cluster (obtained in the HCA in Supplementary Fig. 9) were located closely and that the task organization was similar to that obtained using PCA (Fig. 4).

Comment 4: To formally address whether the patterns fall into clusters (and hierarchical clusters), it would be ideal to inferentially compare cluster (and hierarchical cluster) models to continuous models. For example, one could fit each model to a training set and assess whether the models' predictive performance differs on an independent test set. (This is in contrast to hierarchical cluster analysis, which assumes a hierarchical cluster structure rather than inferring the presence of such a structure from the data.)

Answer: We have constructed a new model that includes information about hierarchical clustering ('Hierarchical model'). We first performed hierarchical cluster analysis using data from five subjects (excluding the target subject's data). This dendrogram (cluster tree) includes 102 non-terminal nodes (the terminal nodes correspond to the 103 tasks). Each of the 102 features of the hierarchical model thus represents whether any tasks subordinated by the target node were performed in each time bin.

We compared the prediction performance of the original task-type model and hierarchical model and found that the hierarchical model outperformed the original task-type model in all subjects. To explain this result, we have added the following text to the Results section:

To further explore whether hierarchical information is useful in capturing cortical representation, we constructed an additional hierarchical model that was based on the task clusters subordinated by each non-terminal node in the dendrogram (see Methods). By comparing the prediction accuracy of brain activity using the task-type model (Fig. 1c) and the hierarchical model, we found that the hierarchical model outperformed the task-type model (hierarchical model, mean \pm SD, 0.313 ± 0.046 ; task-type model, 0.293 ± 0.053 ; Wilcoxon signed-rank tests, $P < 0.001$ for all participants).

We have added the following text to the Methods section:

To further quantify the importance of the estimated hierarchy, we constructed a hierarchical model based on the result of HCA. We first conducted HCA using data from five subjects (excluding the target subject's data). This dendrogram (cluster tree) includes 102 non-terminal nodes (the terminal nodes correspond to the 103 tasks). The feature matrix of the hierarchical model consisted of 102-element binary vectors, which were assigned either 1 or 0 for each time bin, indicating whether any tasks subordinated by the target non-terminal node were performed in that period.

Comment 5: (3) Perform a simple pairwise task decoding analysis

It's great that the decoding analysis generalizes to new tasks. But this requires model-based generalization. It would be useful, additionally, to use decoding to assess the discriminability of the task-related activity patterns in a less model-dependent way.

One could fit a linear discriminant for each pair of tasks and test on independent data from the same subject performing the same two tasks again. (If the accuracy were replaced by the linear discriminant t value or crossnobis estimator, then this could also form the basis for point (2) above.)

Answer: To test the decoding in a less model-dependent way, we have performed an additional decoding analysis directly from brain activity using a support vector machine. The analysis again exhibited high decoding accuracy.

We have therefore added Supplementary Figure 11 and the following text to the Methods section:

To test decoding performance in a model-independent way, we also decoded over 100 tasks directly from brain activity using a support vector machine (Supplementary Fig. 11).

We added the following pieces of text to the Supplementary information:

Supplementary Figure 11. Decoding directly from brain activity. Histogram of decoding accuracies of over 100 tasks obtained using a linear support vector machine for subjects ID01–ID06. The red line indicates the chance-level accuracy (50%). Filled bars indicate tasks that were decoded with significant accuracy (mean decoding accuracy and percentage of significant tasks; ID01, 99.0%, 100%; ID02, 98.1%, 100%; ID03, 98.0%, 100%; ID04, 99.1%, 100%; ID05, 94.5%, 100%; ID06, 98.6%, 100%. Sign tests, $p < 0.05$, FDR-corrected).

Decoding tasks directly from brain activity

To further test whether we can decode over 100 tasks in a less model-dependent way, we also applied a linear support vector machine (SVM) analysis directly to the task-evoked brain activity. We used LIBSVM for the linear SVM decoding of 103-way multi-class classification³². The SVM decoder was constructed based on the response matrix and the task-label vector with an index of 1–103. This analysis exhibited high decoding accuracy for all subjects (mean \pm SD, $97.9 \pm 1.7\%$; all tasks were significant; sign tests, $p < 0.05$, FDR-corrected; Supplementary Fig. 11).

Comment 6: “A cognitive factor encoding model utilizing continuous intermediate features by using metadata-based inferences predicted brain activation patterns for more than 80 % of the cerebral cortex and decoded more than 95 % of tasks, even under novel task conditions.” (abstract)

The numbers 80% and 95% are not meaningful in the absence of additional information (more than 80% of the voxel responses predicted significantly above chance level, and more than 95% of the tasks were significantly distinct from at least some other tasks). You could either add the information needed to interpret these numbers to the abstract or remove the numbers from the abstract. (The abstract should be interpretable in isolation.)

--Nikolaus Kriegeskorte

Answer: We have deleted the numbers 80% and 95% from the abstract and modified the text as follows:

A cognitive factor encoding model utilizing continuous intermediate features by applying metadata-based inferences predicted brain activity and decoded tasks, even under novel task conditions.

Reviewer #3 (Remarks to the Author):

Comment 1: Major comments:

I frankly don't have any major criticisms. The only thing I wonder is if there is something beyond methodology that can be taken away from this paper, and if there is any such takeaway it would be great for the authors to communicate it.

Answer: We believe that constructing a new model and validating it quantitatively are important contributions to the scientific community. From this point of view, the most important finding reported in this article is that we constructed a cognitive factor model that can predict and decode brain activity when the subject is undertaking a novel task. This is a crucial step to developing analyses of abstract cognition using a computational model. Quantitatively, we found that the model achieved significant predictions of

brain activity under novel conditions for approximately 86% of the cortex. This could provide a baseline for the prediction of complete human brain activity under arbitrary cognitive conditions. As the 103 tasks (one of the largest cognitive batteries for single subjects, but still moderate in number) employed here covered a major proportion of the cortex, the number of cognitive factors required to span the entire cognitive space is likely not to be much higher than this.

We have added the following text to the Discussion section:

The fact that our model achieved unprecedentedly wide generalizability regarding cortical coverage and multi-task decodability indicates that our task battery represents a sufficient number of samples capable of probing the major proportion of the human cognitive space and provides a baseline for complete characterization of the cognitive space.

Comment 2: Minor comments:

I found myself having to go to the methods/supplement repeatedly for items that I could imagine would be in the main text/figures; e.g.,

1. Please add a list of many of the tasks to figure 1; all 100 may be too much, but surely 10-20 could fit in a new panel

Answer: In order to show several examples, we have added a new panel to Figure 1 describing 12 tasks. We have added the following text in the caption of Figure 1:

a, Example tasks. Example image of 12 tasks were shown, with task names described at the top. Some visual images used in the tasks are different from the original because of copyright protection.

Comment 3: 2. In the results section it would be useful to have listed the number of subjects and amount of scanning data per subject

Answer: We have added the following text to the Results section:

‘which were concatenated across six subjects (Fig. 2a). The training dataset consisted of 3336 samples (6672 s) and the test dataset consisted of 412 samples (824 s, repeated four times).’

Comment 4: 3. The description of the cognitive factor feature space from Neurosynth in the main text is lacking (e.g., lines 191-195). A reader should be able to understand what you did from that portion alone, and the text there could be improved to clarify what actually occurred.

Answer: To clarify the methodology of the cognitive factor model, we added the following text to the Results section:

The latent feature space was obtained based on the 715 terms and their reverse-inference maps from the Neurosynth database¹³. To produce the cognitive transform function (CTF) for each subject, we calculated the correlation coefficients between the weight map for each task in the task-type model and the reverse-inference map. We then obtained the feature matrix of the cognitive factor model by multiplying the CTF by the feature matrix of the task-type model.

In addition to the changes made in response to the reviewer’s comments, we have made the following changes to our manuscript. These changes did not affect the main conclusion of the study.

*We have relabelled the tasks as follows:

Traffic => TrafficSign

Irony => Sarcasm

RateSexyMovF <=> RateSexyMovM

*We have moved the description of the selection criteria for Neurosynth terms from the ‘Cognitive factor model’ subsection to the ‘Interpretation of cognitive factors related to task clusters’ subsection, which appears earlier in Methods section.

*We noticed that the decoding accuracy in the original submission was based on an older analysis that had a minor misalignment in the task time window. In the revised manuscript, we updated this to the correct version. We have also corrected a bug in the script of FDR correction (a false rejection of one task just under-threshold). Accordingly, the decoding performance was slightly altered in the Results section of the manuscript:

Decoding accuracy: $96.5 \pm 0.9\%$ => $96.0 \pm 0.8\%$;

Percent of significant task: $98.9 \pm 0.4\%$ => $99.5 \pm 0.5\%$

*Minor typographical errors and grammatical mistakes have been corrected.

Reviewers' Comments:

Reviewer #1:

Remarks to the Author:

The authors have tried to address the concerns I raised in the previous manuscript, but on balance I don't feel that these responses have really addressed my fundamental concerns. I appreciate the addition of the hierarchical modeling, but I don't feel that the paper has changed in a way that really changes my view of its potential impact. I also remain concerned about the very high level of classification accuracy, particularly in the context of relatively low correlations (~ 0.3) between predicted and actual images. It seems quite surprising to me that a model that can only explain about 10% of the variance in the actual images could perform almost perfectly when inverted.

The shuffling analysis might address concerns regarding leakage, but insufficient details are provided regarding how this was done. In particular, what was the scope of the shuffling?

I can understand potential administrative impediments to sharing the data, but I think that for a paper like this using custom analysis code, sharing of the code is essential for reviewers and readers to understand what was done.

Reviewer #2:

Remarks to the Author:

The authors have satisfactorily addressed my major concerns.

I think the paper is now ready for publication and will make an inspiring addition to the literature. —
Nikolaus Kriegeskorte

Reviewer #3:

Remarks to the Author:

In my opinion, the authors have done a reasonable job responding to the comments and critiques provided by the reviewers.

I still find the paper to be largely methodological, and I am not clear what scientific inferences can be made. The authors argue that quantitative models are a contribution in and of themselves. I tend to be sympathetic to that kind of argument when: (1) the model is stimulus-computable and therefore suggests one potential means by which the observed responses can be arrived at; and/or (2) the model can be queried to help reveal intuitive explanations of what is occurring. As far as I can tell, the main scientific inference is that of a very coarse parcellation of the cortical sheet into coarse systems ("visual", "memory", "language", "motor", "default mode", and "auditory"). This kind of coarse functional organization is unsurprising -- I imagine it literally is already in textbooks -- as it has been demonstrated by various previous studies.

That said, the paper is methodologically genuinely innovative and opens a new space for subsequent work. So, frankly, I'm a bit equivocal. If the other two reviewers are more enthusiastic than I am, then that would allay some of my concerns.

We are grateful to the reviewers for reviewing our manuscript and providing valuable comments. We have modified our manuscript based on these comments and have provided responses to their comments below. For clarity, our responses are placed after each reviewer comment. Passages that have been newly inserted into the manuscript have been underlined.

Reviewer #1 (Remarks to the Author):

Comment 1: I appreciate the addition of the hierarchical modeling, but I don't feel that the paper has changed in a way that really changes my view of its potential impact.

Answer: Using the current data, we can evaluate the detailed topography of multi-task representation in any cortical voxel for each individual, and such examinations provide novel results that were previously undetectable in single studies. To demonstrate an example, we added additional results on the topographic change of task representation in the inferior parietal lobule (Figure 4e-h; details are described in the response to Reviewer #3's comment).

Comment 2: I also remain concerned about the very high level of classification accuracy, particularly in the context of relatively low correlations (~ 0.3) between predicted and actual images. It seems quite surprising to me that a model that can only explain about 10% of the variance in the actual images could perform almost perfectly when inverted.

Answer: A direct comparison between the mean prediction accuracy and decoding accuracy is not appropriate for the following reasons:

(1) The prediction accuracy was evaluated based on Pearson's correlation coefficient between predicted and actual responses (chance level = 0.0), whereas decoding accuracy was evaluated using one vs. one examinations based on the task likelihood (chance level = 50%, Fig. 6). When we calculated the decoding accuracy using Pearson's correlation coefficient between the actual and decoded features (latent features of cognitive factor model), the decoding accuracy was 0.762 ± 0.019 (mean \pm SD across six participants).

(2) The reported prediction accuracy (~ 0.3) was an average score of the whole cerebral cortex ($\sim 60,000$ voxels, Fig. 5B). This includes regions that tend to have signal loss, such as the orbitofrontal cortex, and these regions may not contribute to the decoding performance.

To evaluate the contribution of informative voxels to the prediction and decoding performance, we selected the top 1000 voxels using the training dataset. For each participant, we randomly divided the

training dataset into training samples (80%) and validation samples (20%) and performed model fitting using the cognitive factor model 50 times. The optimal regularization parameter was determined based on the mean prediction accuracy across the cortex and all repetitions. The top 1000 voxels that exhibited the largest prediction accuracy averaged across 50 repetitions were used in the following analyses. Note that the test dataset was independent of the voxel selection procedure.

When we used only the top 1000 voxels, the mean prediction accuracy of the test dataset (novel tasks) was 0.772 ± 0.039 . The mean decoding accuracy was $95.8\% \pm 1.4\%$ using task likelihood (0.720 ± 0.032 using Pearson's correlation coefficient). Therefore, prediction and decoding analyses produced a comparable performance.

Comment 3: The shuffling analysis might address concerns regarding leakage, but insufficient details are provided regarding how this was done. In particular, what was the scope of the shuffling?

Answer: We agree that the description of shuffling analysis was insufficient. We have therefore added the following content to the Methods section:

“Specifically, all elements of the $[715 \times 412]$ feature matrix of the cognitive factor encoding model in the test dataset were randomly shuffled in an element-wise manner, and predicted responses were calculated by multiplying the shuffled feature matrix with the original weight matrix.”

“Specifically, all elements of the $[103 \times 715]$ CTF matrix were randomly shuffled in an element-wise manner, and the task likelihood was measured using Pearson's correlation coefficient between the shuffled CTF of each task and each decoded cognitive factor vector.”

Comment 4: I can understand potential administrative impediments to sharing the data, but I think that for a paper like this using custom analysis code, sharing of the code is essential for reviewers and readers to understand what was done.

Answer: We clarified institutional administrative obstacles for data and code sharing. We uploaded raw MRI data on OpenNeuro.org (<https://openneuro.org/datasets/ds002306>; Currently, data can be viewed only by the shared users. Please inform your email address if you want to have an access). We also shared the code and preprocessed data in the Open Science Framework (OSF)

(https://osf.io/ea2jc/?view_only=9d05bf520cdc475eb8f9c3295d8c2d72). At present, these materials are open only for reviewing purposes, and we will make the data and codes publicly available after the acceptance of this manuscript.

We therefore updated our Code and Data availability statement in the Methods section as follows:

“The MATLAB code used in the present study and the datasets generated and/or analysed during the current study are available at the Open Science Framework (OSF, <https://osf.io/ea2jc/>). The source data underlying all Figures, Supplementary Figures, and Tables except Figure 1 are also provided as a Source Data file at the OSF repository. The raw MRI data are available at the OpenNeuro.org (<https://openneuro.org/datasets/ds002306>).”

Reviewer #3 (Remarks to the Author):

Comment 1: I still find the paper to be largely methodological, and I am not clear what scientific inferences can be made. The authors argue that quantitative models are a contribution in and of themselves.

I tend to be sympathetic to that kind of argument when: (1) the model is stimulus-computable and therefore suggests one potential means by which the observed responses can be arrived at; and/or (2) the model can be queried to help reveal intuitive explanations of what is occurring.

As far as I can tell, the main scientific inference is that of a very coarse parcellation of the cortical sheet into coarse systems ("visual", "memory", "language", "motor", "default mode", and "auditory"). This kind of coarse functional organization is unsurprising -- I imagine it literally is already in textbooks -- as it has been demonstrated by various previous studies.

That said, the paper is methodologically genuinely innovative and opens a new space for subsequent work. So, frankly, I'm a bit equivocal. If the other two reviewers are more enthusiastic than I am, then that would allay some of my concerns.

Answer: We showed example clusters of ("visual", "memory", "language", "motor", "introspection", and "auditory") because these were the largest clusters to summarize the results. However, we can scrutinize any subclusters using the same analysis.

For example, there is a time-perception subcluster (“TimeSound”, “Rhythm”) within the Auditory cluster. The reverse-inference analysis assigned cognitive factors of “timing”, “monitoring”, and “working memory” to this subcluster in addition to the auditory factors.

In the knowledge-recalling subcluster (“RecallKnowledge”, “CategoryFluency”) within the Introspection cluster, the reverse inference analysis assigned cognitive factors of “phonological”, “production”, and “language” even though the participants were not asked to overtly reveal linguistic information.

These are examples of non-trivial findings, which were newly revealed by combining hierarchical clustering analysis and the reverse-inference method.

To describe the above results, we have added Supplementary Table 1 and added the following content to the Results section:

The reverse-inference analysis is applicable to any subcluster (Supplementary Table 1). For example, there is a time-perception subcluster (“TimeSound”, “Rhythm”) within the Auditory cluster. The reverse-inference analysis assigned cognitive factors of “timing”, “monitoring”, and “working memory” to this subcluster in addition to the auditory factors. In the knowledge-recalling subcluster (“RecallKnowledge”, “CategoryFluency”) within the Introspection cluster, the reverse-inference analysis assigned cognitive factors of “phonological”, “production”, and “language” even though the participants were not asked to overtly produce speech.

In addition to the clustering analysis, we can also scrutinize the representations of cortical voxels. This is particularly important when we study the association cortex, which has been linked to many cognitive domains. For example, studies have reported that the inferior parietal lobule (IPL) is associated with number processing (Harvey et al. Science 2013), visuospatial attention (Park et al. Hum. Brain Mapp. 2016), and sensorimotor processing (Naito et al., Cereb. cortex 2017). When we examined the task representation in the left IPL, we found a topographic change of task selectivity along with the inferior to superior direction (Figure 4e-h). Calculation and logical inference tasks were represented in all three voxels (Figure 4f-h), whereas motor tasks were largely represented in the inferior voxel (Figure 4f) and visual tasks were largely represented in the superior voxel (Figure 4h). The middle voxel showed an intermediate representation for both motor and visual tasks (Figure 4g). Such topographic changes in task selectivity are indicated by the cortical map (Figure 4a, e), which displays a color change from red (inferior voxel) to black (middle voxel) to green (superior voxel).

To convey these results, we have added Figure 4e-h and have added the following content to the figure legend of Figure 4:

e, The left inferior parietal lobule (IPL) corresponding to the orange square part in the inflated cortical sheet (a). Three voxels along with the inferior to superior direction were selected to show the topographic change of task selectivity (f-h). For better visibility, only 18 tasks with positive weight values are shown in white.

We have also added the following content to the Results section:

To examine how each cortical voxel differs in its representation of over 100 tasks (*task selectivity*), we visualized voxel-wise task weights on the two-dimensional cognitive space depicted in Figure 3a. We found a representation of language-related tasks in the middle temporal voxel (Figure 4b), introspection-related tasks in the left medial frontal voxel (Figure 4c), and auditory-related tasks in the right superior temporal voxel (Figure 4d). Using this visualization method, scrutiny of fine mappings of task selectivity of any cortical voxel is possible. For example, when we examined the left inferior parietal lobule (IPL), we found a topographic change of task selectivity along with the inferior to superior direction (Figure 4e-h). Calculation and logical inference tasks were represented in all three voxels (Figure 4f-h), whereas motor tasks were largely represented in the inferior voxel (Figure 4f), and visual tasks were largely represented in the superior voxel (Figure 4h). The middle voxel showed an intermediate representation for both motor and visual tasks (Figure 4g). Such topographic changes of task selectivity are indicated by the cortical map (Figure 4a, e), which displays a color change from red (inferior voxel) to black (middle voxel) to green (superior voxel). These results suggest that voxel-wise modeling can entangle the complex topography of multiple cognitive dimensions in the association cortex.

We have added the following content to the Discussion section:

This is particularly effective in association cortices, such as the IPL, which has been associated with various cognitive domains²².

We have cited the following reference in the References section:

22. Humphreys, G. F. & Lambon Ralph, M. A. Fusion and fission of cognitive functions in the human parietal cortex. *Cereb. Cortex* 25, 3547–3560 (2015).

As we mentioned in our response to Reviewer #1, we will make our raw MRI data publicly available in OpenNeuro.org (<https://openneuro.org/datasets/ds002306>; Currently, data can be viewed only by the shared users. Please inform your email address if you want to have an access.) and our code and preprocessed data in OSF (https://osf.io/ea2jc/?view_only=9d05bf520cdc475eb8f9c3295d8c2d72).

Although we showed several representative examples in the manuscript, those interested in the cognitive functions covered in the current study can examine the organization of subclusters and task selectivity in cortical voxels. We believe that our manuscript and the accompanying open data will contribute to a more comprehensive understanding of human cognitive functions.

Reviewers' Comments:

Reviewer #3:

Remarks to the Author:

The authors have made efforts to address the concerns raised in previous reviews -- I particularly appreciate their expanded analyses of finer structure beyond the coarse parcellation. I believe the paper will make a solid contribution.